# Learning non-Markovian Decision-Making from State-only Sequences

**Aoyang Qin**[*,1,2]  **Feng Gao**[3]  **Qing Li**[2]  **Song-Chun Zhu**[1,2,4]  **Sirui Xie**[*,5]

[1] Department of Automation, Tsinghua University
[2] Beijing Institute for General Artificial Intelligence (BIGAI)
[3] Department of Statistics, UCLA [4] School of Artificial Intelligence, Peking University
[5] Department of Computer Science, UCLA

## Abstract

Conventional imitation learning assumes access to the actions of demonstrators, but these motor signals are often non-observable in naturalistic settings. Additionally, sequential decision-making behaviors in these settings can deviate from the assumptions of a standard Markov Decision Process (MDP). To address these challenges, we explore deep generative modeling of state-only sequences with non-Markov Decision Process (nMDP), where the policy is an energy-based prior in the latent space of the state transition generator. We develop maximum likelihood estimation to learn both the transition and the policy, which involves short-run MCMC sampling from the prior and importance sampling for the posterior. The learned model enables *decision-making as inference*: model-free policy execution is equivalent to prior sampling, model-based planning is posterior sampling initialized from the policy. We demonstrate the efficacy of the proposed method in a prototypical path planning task with non-Markovian constraints and show that the learned model exhibits strong performances in challenging domains from the MuJoCo suite.

## 1 Introduction

Imitation from others is a prevalent phenomenon in humans and many other species, where individuals learn by observing and mimicking the actions of others. An intriguing aspect of this process is the brain's ability to extract motor signals from sensory input. This remarkable capability is facilitated by *mirror neurons* [1, 2], which respond to observations as if the imitator is performing the actions themselves. In conventional imitation learning [3, 4] and offline reinforcement learning [5], action labels have served as proxies for mirror neurons. But it is important to recognize that they are actually productions of human interventions. Given the recent advancements in AI, now is probably an opportune time to explore imitation learning in a more naturalistic setting.

While the setting of state-only demonstrations is not common, there are certain exceptions. For example, Inverse Reinforcement Learning (IRL) initially formulated the problem as state visitation matching [6], where demonstrations consist solely of state sequences. Subsequently, this state-only setting was rebranded as Imitation Learning from Observations (ILfO), which introduced the generalized formulation of matching marginal state distributions [7, 8]. These methods typically rely on the Markov assumption and Temporal Difference (TD) learning techniques [9]. One consequence of this assumption, previously believed to be advantageous, is that sequences with different state orders are treated as equivalent. However, the success of general sequence modeling [10] has challenged

---

[*] indicates equal contribution. Correspondence: Sirui Xie (srxie@ucla.edu). Code and data are available at
https://github.com/qayqaq/LanMDP

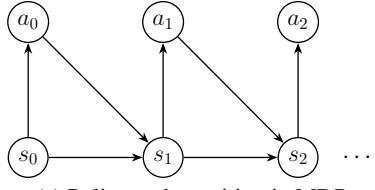
(a) Policy and transition in MDP

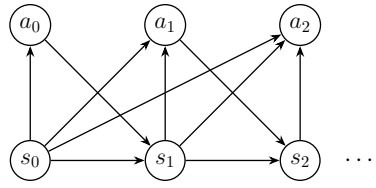
(b) Policy and transition in nMDP

Figure 1: **Graphical model of policy and transition** in standard Markov Decision Process and non-Markov Decision Process. Reward variables are omitted in the probabilistic graph to emphasize the difference in dependency between actions and states. nMDP is a natural generalization of standard MDP.

this belief, leading to deep reflections. Notable progresses since then include an analysis of the expressivity of Markovian rewards [11] and a series of sequence models tailored for decision-making problems [12–15]. Aligning with this evolving trend, we extend the state-only imitation learning problem to encompass non-Markovian domains.

In this work, we propose a generative model based on non-Markov Decision Process (nMDP), in which states are fully observable and actions are latent. Unlike existing monolithic sequence models, we factorize the joint state-action distribution into policy and causal transition according to the standard Markov Decision Process (MDP). To further extend to the non-Markovian domain, we condition the policy on sequential contexts. The density families of policy and transition are consistent with conventional IRL [4]. We refer to this model as Latent-action non-Markov Decision Process (LanMDP). Because the actions are latent variables following Boltzmann distribution, the present model is closely related to the Latent-space Energy-Based Model (LEBM) [16]. To learn the latent policy by Maximum Likelihood Estimation (MLE), we need to sample from the prior and the posterior. We sample the prior using short-run Markov Chain Monte Carlo (MCMC) [17], and the posterior using importance sampling. Specifically, the proposed importance sampling sidesteps back-propagation through time in posterior MCMC with a single-step lookahead of the Markov transition. The transition is learned from self-interaction.

Once the LanMDP is learned, it can be used for policy execution and planning through prior and posterior sampling, or in other words, *policy as prior, planning as posterior inference* [18, 19]. In our analysis, we derive an objective of the non-Markovian decision-making problem induced from the MLE. We show that the prior sampling at each step can indeed lead to optimal expected returns. Almost surprisingly, we find that the entire family of maximum entropy reinforcement learning [4, 20–24] naturally emerges from the algebraic structures in the MLE of latent policies. This formulation avoids the peculiarities of maximizing state transition entropy in prior arts [20, 24]. We also show that when a target goal state is in-distribution, the posterior sampling is optimizing a conditional variant of the objective, realizing model-based planning. In our experiments, we validate the necessity and efficacy of our model in learning to sequentially plan cubic curves, and illustrate an *over-imitation* phenomenon [25, 26] when the learned model is repurposed for goal-reaching. We also test the proposed modeling, learning, and computing method in MuJoCo, a domain with higher-dimensional state and action spaces, and achieve performance competitive to existing methods, even those that learn with action labels.

## 2 Non-Markov Decision Process

The most well-known sequence model of a decision-making process is Markov Decision Process. A MDP is a tuple $\mathcal{M} = \langle S, A, T, R, \rho, H \rangle$ that contains a set $S$ of states, a set $A$ of actions, a transition $T : S \times A \mapsto \Pi(S)$ that returns for every state $s_t$ and action $a_t$ a distribution over the next state $s_{t+1}$; a reward function $R : S \times A \mapsto \mathbb{R}$ that specifies the real-valued reward received by the agent when taking action $a_t$ in state $s_t$; an initial state distribution $\rho : \Pi(S)$; and a horizon $H$ that is the maximum number of actions/steps the agent can execute in one episode. A solution to an MDP is a policy that maps states to actions, $\pi : S \mapsto \Pi(A)$. The value of policy $\pi$, $V^\pi(s) = \mathbb{E}_{T,\pi}[\sum_{t=0}^{H} R(s_t)|s_0 = s]$ is the expected cumulative reward (*i.e.* return) when executing with this policy starting from state $s$. The state-action value of policy $\pi$ is $Q^\pi(s_t, a_t) = R(s_t, a_t) + \mathbb{E}_{T(s_{t+1}|s_t,a_t)}[V^\pi(s_{t+1})]$. The optimal policy $\pi^*$ can maximize either $E_{\rho(s_0)}[V^\pi(s_0)]$, or the same objective plus the policy entropy [27, 4, 22]. The Markovian assumption supports the convergence of a series of TD-learning methods [9], whose reliability in non-Markovian domains is still an open problem.

A non-Markov Decision Process is also a tuple $\mathcal{M} = \langle S, A, T, R, \rho, H \rangle$. It generalizes MDP by allowing for non-Markovian transitions and rewards [28]. Notably, assuming Markovian transition and non-Markovian reward is usually sufficient since a state space with non-Markovian transition can be represented with its Markov abstraction [29]. Markov abstraction can be done either by treating the original space as observations generated from the latent belief state in a Partially Observable Markov Decision Process (POMDP) [30], or by projecting historic contexts into an embedding space for sequence pattern detection [31, 32, 28]. Presumably, it is statistically more interesting in deep learning to focus our attention on non-Markovian domains where the temporal dependencies in transition and reward differ. Therefore, without loss of generality, we assume that the state transition is Markovian $T : S \times A \mapsto \Pi(S)$, while the reward is not [33, 11], *i.e.* $R : S^+ \mapsto \mathbb{R}$, with $S^+$ denotes the set of all finite non-empty state sequences with length smaller than $H$. Obviously, the policy should also be non-Markovian $\pi : S^+ \mapsto \Pi(A)$. Check Figure 1 for a probabilistic graphical model of the generation process of state sequences from a policy.

## 3 Learning and Sampling

### 3.1 Latent-action nMDP

A complete trajectory is denoted by

$$\zeta = \{s_0, a_0, s_1, a_1, \cdots, a_{T-1}, s_T\}, \tag{1}$$

where $T$ is the maximum length of all observed trajectories and $T \leqslant H$. The joint distribution of state and action sequences can be factorized according to the causal assumptions in nMDP:

$$p_\theta(\zeta) = p(s_0)p_\alpha(a_0|s_0)p_\beta(s_1|s_0, a_0) \cdots p_\alpha(a_{T-1}|s_{0:T-1})p_\beta(s_T|s_{T-1}, a_{T-1})$$
$$= p(s_0) \prod_{t=0}^{T-1} p_\alpha(a_t|s_{0:t})p_\beta(s_{t+1}|s_t, a_t), \tag{2}$$

where $p_\alpha(a_t|s_{0:t-1})$ is the policy model with parameter $\alpha$, $p_\beta(s_t|s_{t-1}, a_{t-1})$ is the transition model with parameter $\beta$, both of which are parameterized with neural networks, $\theta = (\alpha, \beta)$. $p(s_0)$ is the initial state distribution, which can be sampled as a black box.

The density families of policy and transition are consistent with the conventional setting of IRL [4], where the transition describes the predictable change in state as a single-mode Gaussian, $s_{t+1} \sim \mathcal{N}(g_\beta(s_t, a_t), \sigma^2)$, and the policy accounts for bounded rationality as a Boltzmann distribution with state-action value as the unnormalized energy:

$$p_\alpha(a_t|s_{0:t}) = \frac{1}{Z(\alpha, s_{0:t})} \exp\left(f_\alpha(a_t; s_{0:t})\right), \tag{3}$$

where $f_\alpha(a_t; s_{0:t})$ is the negative energy, $Z(\alpha, s_{0:t}) = \int \exp(f_\alpha(a_t; s_{0:t}))da_t$ is the normalizing constant given the contexts $s_{0:t}$. We discuss a general push-forward transition in Appx A.3.

Since we can only observe state sequences, the aforementioned generative model can be understood as a sequential variant of LEBM [16], where the transition serves as the generator and the policy is a history-conditioned latent prior. The marginal distribution of state sequences and the posterior distribution of action sequences are:

$$p_\theta(s_{0:T}) = \int p_\theta(s_{0:T}, a_{0:T-1})da_{0:T-1}, \quad p_\theta(a_{0:T-1}|s_{0:T}) = \frac{p_\theta(s_{0:T}, a_{0:T-1})}{p_\theta(s_{0:T})}. \tag{4}$$

### 3.2 Maximum likelihood learning

We need to estimate $\theta = (\alpha, \beta)$. Suppose we observe *offline* training examples: $\{\xi^i\}, i = 1, 2, \cdots, n, \quad \xi^i = [s_0^i, s_1^i, ..., s_T^i]$. The log-likelihood function is:

$$L_{off}(\theta) = \sum_{i=1}^{n} \log p_\theta(\xi^i). \tag{5}$$

Denote posterior distribution of action sequence $p_\theta(a_{0:T-1}|s_{0:T})$ as $p_\theta(A|S)$ for convenience where $A$ and $S$ means the complete action and state sequences in a trajectory. The full derivation of the learning method can be found in Appx A.2, which results in the following gradient:

$$\nabla_\theta \log p_\theta(\xi) = \mathbb{E}_{p_\theta(A|S)}[\sum_{t=0}^{T-1} (\underbrace{\nabla_\alpha \log p_\alpha(a_t|s_{0:t})}_{\text{policy/prior}}, \underbrace{\nabla_\beta \log p_\beta(s_{t+1}|s_t, a_t)}_{\text{transition}})]. \tag{6}$$

Due to the normalizing constant $Z(\alpha, s_{0:t})$ in the energy-based prior $p_\alpha$, the gradient for the policy term involves both posterior and prior samples:

$$\delta_{\alpha,t}(S) = \mathbb{E}_{p_\theta(A|S)}\left[\nabla_\alpha \log p_\alpha(a_t|s_{0:t})\right] = \mathbb{E}_{p_\theta(A|S)}\left[\nabla_\alpha f_\alpha(a_t; s_{0:t})\right] - \mathbb{E}_{p_\alpha(a_t|s_{0:t})}\left[\nabla_\alpha f_\alpha(a_t; s_{0:t})\right], \tag{7}$$

where $\delta_{\alpha,t}(S)$ denotes the expected gradient of policy term for time step $t$. Intuition can be gained from the perspective of adversarial training [34, 35]: On one hand, the model utilizes action samples from the posterior $p_\theta(A|S)$ as pseudo-labels to supervise the unnormalized prior at each step. On the other hand, it discourages action samples directly sampled from the prior. The model converges when prior samples and posterior samples are indistinguishable.

To ensure the transition model's validity, it needs to be grounded in real-world dynamics $Tr$ when jointly learned with the policy. Otherwise, the agent would be purely hallucinating based on the demonstrations. Throughout the training process, we allow the agent to periodically collect *on-policy* data $\{(s_t^i, a_t^i, s_{t+1}^i)\}$, $i = 1, 2, \cdots, m$, $t = 1, 2, \cdots, T$ with $p_\alpha(a_t|s_{0:t})$ and update the transition with a *composite likelihood* [36]

$$L_{comp}(\beta) = L_{off}(\theta) + L_{on}(\beta), \quad L_{on}(\beta) = \sum_{i=1}^m \sum_{t=1}^T \log p_\beta(s_{t+1}^i|s_t^i, a_t^i). \tag{8}$$

## 3.3 Prior and posterior sampling

The maximum likelihood estimation requires samples from the prior and the posterior distributions of actions. It would not be a problem if the action space is quantized. However, since we target general latent action learning, we proceed to introduce sampling techniques for continuous actions.

When sampling from a continuous energy space, short-run Langevin dynamics [17] can be an efficient choice. For a target distribution $\pi(a)$, Langevin dynamics iterates $a_{k+1} = a_k + s\nabla_{a_k} \log \pi(a_k) + \sqrt{2s}\epsilon_k$, where $k$ indexes the number of iteration, $s$ is a small step size, and $\epsilon_k$ is the Gaussian white noise. $\pi(a)$ can be either the prior $p_\alpha(a_t|s_{0:t})$ or the posterior $p_\theta(A|S)$. One property of Langevin dynamics that is particularly amenable for EBM is that we can get rid of the normalizing constant. So for each $t$ the iterative update for prior samples is

$$a_{t,k+1} = a_{t,k} + s\nabla_{a_{t,k}} f_\alpha(a_{t,k}; s_{0:t}) + \sqrt{2s}\epsilon_k. \tag{9}$$

Given a state sequence $s_{0:T}$ from the demonstrations, the posterior samples at each time step $a_t$ come from the conditional distribution $p(a_t|s_{0:T})$. Notice that with Markov transition, we can derive

$$p_\theta(a_{0:T-1}|s_{0:T}) = \prod_{t=0}^{T-1} p_\theta(a_t|s_{0:T}) = \prod_{t=0}^{T-1} p_\theta(a_t|s_{0:t+1}). \tag{10}$$

Eq. (10) reveals that given the previous and the next subsequent state, the posterior can be sampled at each step independently. So the posterior iterative update is

$$a_{t,k+1} = a_{t,k} + s\nabla_{a_{t,k}} (\underbrace{\log p_\alpha(a_{t,k}|s_{0:t})}_{\text{policy/prior}} + \underbrace{\log p_\beta(s_{t+1}|s_t, a_{t,k})}_{\text{transition}}) + \sqrt{2s}\epsilon_k. \tag{11}$$

Intuitively, action samples at each step are updated by back-propagation from its prior energy and a single-step lookahead. While gradients from the transition term are analogous to the inverse dynamics in Behavior Cloning from Observations (BCO) [37], it may lead to poor training performance due to non-injectiveness in forward dynamics [38].

We develop an alternative posterior sampling method with importance sampling to overcome this challenge. Leveraging the learned transition, we have

$$p_\theta(a_t|s_{0:t+1}) = \frac{p_\beta(s_{t+1}|s_t, a_t)}{\mathbb{E}_{p_\alpha(a_t|s_{0:t})}\left[p_\beta(s_{t+1}|s_t, a_t)\right]} p_\alpha(a_t|s_{0:t}). \tag{12}$$

Let $c(a_t; s_{0:t+1}) = \mathbb{E}_{p_\alpha(a_t|s_{0:t})}\left[p_\beta(s_{t+1}|s_t, a_t)\right]$, posterior sampling from $p_\theta(a_{0:T-1}|s_{0:T})$ can be realized by adjusting importance weights of independent samples from the prior $p_\alpha(a_t|s_{0:t})$, in which the estimation of weights involves another prior sampling. In this way, we avoid back-propagating through non-injective dynamics and save some computation overhead.

To train the policy, Eq. (7) can now be rewritten as

$$\delta_{\alpha,t}(S) = \mathbb{E}_{p_\alpha(a_t|s_{0:t})}\left[\frac{p_\beta(s_{t+1}|s_t, a_t)}{c(a_t; s_{0:t+1})} \nabla_\alpha f_\alpha(a_t; s_{0:t})\right] - \mathbb{E}_{p_\alpha(a_t|s_{0:t})}\left[\nabla_\alpha f_\alpha(a_t; s_{0:t})\right]. \tag{13}$$

# 4 Decision-making as Inference

In Section 3, we present our method within the framework of probabilistic inference, providing a self-contained description. However, from a decision-making perspective, the learned policy may appear arbitrary. In this section, we establish a connection between probabilistic inference and decision-making, contributing a novel analysis that incorporates the latent action setting, the non-Markovian assumption, and maximum likelihood learning. This analysis is inspired by, but distinct from, previous studies on the relationship between these two fields [39, 20, 40, 41, 24].

## 4.1 Policy execution with prior sampling

Let the ground-truth distribution of demonstrations be $p^*(s_{0:T})$, and the learned marginal distributions of state sequences be $p_\theta(s_{0:T})$. Eq. (5) in Section 3.2 is an empirical estimate of

$$\mathbb{E}_{p^*(s_{0:T})}[\log p_\theta(s_{0:T})] = \mathbb{E}_{p^*(s_0)}\left[\log p^*(s_0) + \mathbb{E}_{p^*(s_{1:T}|s_0)}[\log p_\theta(s_{1:T}|s_0)]\right]. \tag{14}$$

We can show that a sequential decision-making problem can be constructed to maximize the same objective. Our main result is summarized as Theorem 1.

**Theorem 1.** *Assuming the Markovian transition $p_{\beta*}(s_{t+1}|s_t, a_t)$ is known, the ground-truth conditional state distribution $p^*(s_{t+1}|s_{0:t})$ for demonstration sequences is accessible, we can construct a sequential decision-making problem, based on a reward function $r_\alpha(s_{t+1}, s_{0:t}) := \log \int p_\alpha(a_t|s_{0:t})p_{\beta*}(s_{t+1}|s_t, a_t)da_t$ for an arbitrary energy-based policy $p_\alpha(a_t|s_{0:t})$. Its objective is*

$$\sum_{t=0}^{T} \mathbb{E}_{p^*(s_{0:t})}[V^{p_\alpha}(s_{0:t})] = \mathbb{E}_{p^*(s_{0:T})}\left[\sum_{t=0}^{T}\sum_{k=t}^{T} r_\alpha(s_{k+1}; s_{0:k})\right],$$

*where $V^{p_\alpha}(s_{0:t}) := E_{p^*(s_{t+1:T}|s_{0:t})}[\sum_{k=t}^{T} r_\alpha(s_{k+1}; s_{0:k})]$ is the value function for $p_\alpha$. This objective yields the same optimal policy as the Maximum Likelihood Estimation $\mathbb{E}_{p^*(s_{0:T})}[\log p_\theta(s_{0:T})]$.*

*If we further define a reward function $r_\alpha(s_{t+1}, a_t, s_{0:t}) := r_\alpha(s_{t+1}, s_{0:t}) + \log p_\alpha(a_t|s_{0:t})$ to construct a Q function for $p_\alpha$*

$$Q^{p_\alpha}(a_t; s_{0:t}) := \mathbb{E}_{p^*(s_{t+1}|s_{0:t})}\left[r_\alpha(s_{t+1}, a_t, s_{0:t}) + V^{p_\alpha}(s_{0:t+1})\right].$$

*The expected return of $Q^{p_\alpha}(a_t; s_{0:t})$ forms an alternative objective*

$$\mathbb{E}_{p_\alpha(a_t|s_{0:t})}[Q^{p_\alpha}(a_t; s_{0:t})] = V^{p_\alpha}(s_{0:t}) - \mathcal{H}_\alpha(a_t|s_{0:t}) - \sum_{k=t+1}^{T-1} \mathbb{E}_{p^*(s_{t+1:k}|s_{0:t})}[\mathcal{H}_\alpha(a_k|s_{0:k})]$$

*that yields the same optimal policy, for which the optimal $Q^*(a_t; s_{0:t})$ can be the energy function.*

*Only under certain conditions, this sequential decision-making problem is solvable through non-Markovian extensions of the maximum entropy reinforcement learning algorithms.*

*Proof.* See Appx B. $\qquad\square$

The constructive proof above offers profound insights. By starting with the hypothesis of latent actions and MLE, and then considering known transition and accessible ground-truth conditional state distribution, we witness the *automatic emergence* of the entire family of maximum entropy (inverse) RL. This includes prominent algorithms such as soft policy iteration [20], soft Q learning [22] and soft Actor-Critic (SAC) [23]. Among them, SAC is the best-performing off-policy RL algorithm in practice. Unlike the formulation with joint state-action distribution [20, 24], our formulation avoids the peculiarities associated with maximizing state transition entropy. The choice of the maximum entropy policy aligns naturally with the objective of capturing uncertainty in latent actions, and it offers inherent advantages for exploration in model-free learning [42, 22].

## 4.2 Model-based planning with posterior sampling

Lastly, with the learned model, we can do posterior sampling given any complete or incomplete state sequences. The computation involved is analogous to model-based planning. In Section 3.3, we introduce posterior sampling with short-run MCMC and importance sampling when we have the target next state, which generalizes all cases where the targets of immediate subsequent states are given. Here we introduce the complementary case, where the goal state $s_T$ is given as the target.

The posterior of actions given the sequential context $s_{0:t}$ and a target goal state $s_T$ is

$$p_\theta(a_{t:T}|s_{0:t}, s_T) \propto p_\theta(a_{t:T}, s_T|s_{0:t})$$
$$= \int \prod_{k=0}^{T-t-1} [p_\beta(s_{t+k+1}|a_{t+k}, s_{t+k})p_\alpha(a_{t+k}|s_{0:t+k})] \, p_\beta(s_T|a_{T-1}, s_{T-1})ds_{t+1:T-1}, \quad (15)$$

in which all Gaussian expectation $\mathbb{E}_{p_\beta}[\cdot]$ can be approximated with the mean [43]. Therefore, $a_{t:T}$ can be sampled via short-run MCMC with $\nabla_{a_{t:T}} \log p_\theta(a_{t:T}, s_T|s_{0:t})$ back propagated through time. The learned prior can be used to initialize these samples and facilitate the MCMC mixing.

# 5 Experiments

## 5.1 Cubic curve planning

To demonstrate the necessity of non-Markovian value and test the efficacy of the proposed model, we designed a motivating experiment. Path planning is a prototypical decision-making problem, in which actions are taken in a 2D space, with the x-y coordinates as states. To simplify the problem without loss of generality, we can further assume $x_t$ to change with constant speed $h$, such that the action is $\Delta y_t$. Obviously, the transition model $(x_{t+1}, y_{t+1}) = (x_t + h, y_t + \Delta y_t)$ is Markovian.

Path planning can have various objectives. Imagining you are a passenger of an autonomous driving vehicle. You would not only care about whether the vehicle reaches the goal without collision but also how comfortable you feel. To obtain comforting smoothness and curvature, consider $y$ is constrained to be a cubic polynomial $F(x) = ax^3 + bx^2 + cx + d$ of $x$, where $(a, b, c, d)$ are polynomial coefficients. Then the policy for this decision-making problem is non-Markovian.

To see that, suppose we are at $(x_t, y_t)$ at this moment, and the next state should be $(x_t + h, F(x_t + h))$. With Taylor expansion, we know $F(x_t + h) \approx F(x_t) + F'(x_t)h + \frac{F''(x_t)}{2!}h^2 + \frac{F'''(x_t)}{3!}h^3$, so we can have a representation for the policy, $\pi(\Delta y_t|x_t, y_t) = F'(x_t)h + \frac{F''(x_t)}{2!}h^2 + \frac{F'''(x_t)}{3!}h^3$. However, our representation of state only gives us $(x_t, y_t)$, so we will need to estimate those derivatives. This can be done with the finite difference method if we happen to remember the previous states $(x_{t-1}, y_{t-1})$, ..., $(x_{t-3}, y_{t-3})$. Taking the highest order derivative for example, $F'''(x_t) = (y_t - 3y_{t-1} + 3y_{t-2} - y_{t-3})/h^3$. It is thus apparent that the policy would not be possibly represented if we are Markovian or don't remember sufficiently many prior states.

This representation of policy is what models should learn through imitation. However, they should not know the polynomial structure a priori. Given a sufficient number of demonstrations with different combinations of polynomial coefficients, models are expected to discover this rule by themselves. This experiment is a minimum viable prototype for general non-Markovian decision-making. It can be easily extended to higher-order and higher-dimensional state sequences.

**Setting** We employ multi-layer perception (MLP) for this experiment. Demonstrations can be generated by rejection sampling. We constrain the demonstration trajectories to the $(x, y) \in (-1, 1) \times (-1, 1)$ area, and randomly select $y$ and $y'$ at $x = -1$ and $x = 1$. Curves with third-order coefficients less than 1 are rejected. Otherwise, the models may be confused in learning the cubic characteristics.

Non-Markovian dependency and latent energy-based policy are two prominent features of the proposed model. To test the causal role of non-Markovianness, we experiment with context length $\{1, 2, 4, 6\}$. Context length refers to how many prior states the policy is conditioned on. When it is 1, the policy is Markovian. From our analysis above, we know that context length 4 should be the ground truth, which helps categorize context lengths 2 and 6 into insufficient and excessive expressivity. With these four context lengths, we also train Behavior Cloning (BC) models as the control group. In a deterministic environment, there should not be a difference between BC and BCO, as the latter basically employs inverse dynamics to recover action labels. For our model, this simple transition can either be learned or implanted. Empirically, we don't notice a significant difference.

Performance is evaluated both qualitatively and quantitatively. As a 2D planning task, a visualization of the planned curves says a thousand words. In our experiment, we take $h = 0.1$, so the planned paths are rather discretized. We use mean squared error to fit a cubic polynomial and use the residual error as a metric. When calculating the residual error, we exclude those with a third-order coefficient is less than 0.5. Actually, the acceptance rate itself is also a viable metric. It is the number of accepted

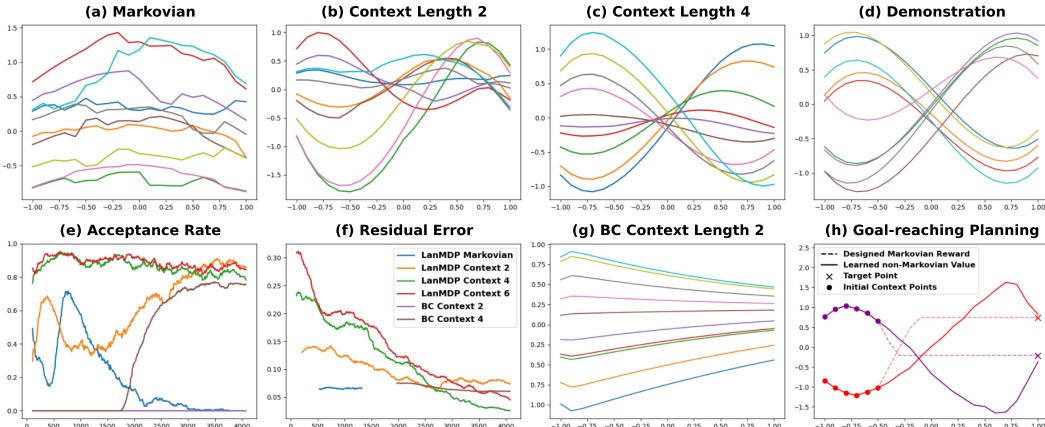

Figure 2: **Results for cubic curve generation**. (a-c) show curves generated at training step 3000 with context lengths 1, 2, 4. Starting points are randomly selected, and all following are sampled from the policy model. Only models with context length 4 learn the cubic characteristic. (d) shows curves from demonstrations. (e) and (f) present the smoothed acceptance rate and fitting residual of trajectories from policies with context lengths 1, 2, 4, 6. The x-axis is the training steps. (e)(f) are better to be viewed together because residual errors will only be calculated if the acceptance rate is above a threshold. For context length 1, the acceptance rate is always zero for BC, so it is not plotted here. (g) shows curves planned by BC with context length 2. It can be compared with (b). Interestingly, LanMDP with context length 2 demonstrates certain cubic characteristics when trained sufficiently long, while the BC counterpart only plans straight lines. (h) is the result of goal-reaching planning, where the dashed line comes from a hand-designed Markov reward, the solid line from the trained LanMDP.

trajectories divided by the total number of testing trajectories. It is complementary to the residual error because it directly measures the understanding of cubic polynomials.

**Results**  Fig. 2(a-c) show paths generated with LanMDP after training for 3000 steps. They have context lengths 1, 2, 4 respectively. Compared with demonstrations in Fig. 2(d), only paths from the policy with context length 4 exhibit cubic characteristics. The Markovian policy totally fails this task. But it still generates curves, rather than straight lines from Markovian BC (see Fig. A1). The policy with context length 2 can plan cubic-like curves at times. But some of its generated paths are very different from demonstrations. To investigate this interesting phenomenon, we plot the training curves in Fig. 2(e)(f). While LanMDP policies with sufficient and excessive expressivity achieve high acceptance rates at the very beginning of the training, policies with Markovian and insufficient expressivity struggle to generate expected curves at the same time. Remarkably, as training goes by, the policy with context length 2, which can only approximate the ground-truth action in the first order, gradually improves in acceptance rate and residual error. This observation is consistent with Fig. 2(b).

Continuing our investigation, we plot curves generated by its BC counterparts in Fig. 2(g) but only see straight lines like the Markovian BC. Therefore, we conjecture that the LanMDP policy with length context 2 leverages its energy-based multi-modality to capture the uncertainty induced by marginalizing part of the necessary contexts. The second-order error in Taylor expansion is possibly remedied by this, especially after long-run training. The Markovian LanMDP policy, however, fails to unlock such potential because it cannot even figure out the first-order derivative.

There are some other note-worthy observations. (i) Excessive expressivity does not impair performance, it just requires more training. As shown in  Fig. 2(e)(f), at the end of training, LanMDP policies with context length 6 perform as well as ones with context length 4. This demonstrates LanMDP's potential in inducing proper state abstraction from sequential contexts. TD learning, however, has been shown to be incapable of such abstraction in a prior work [44]. (ii) BC policies with sufficient contexts do not perform as well as LanMDP, as shown in Fig. 2(e)(f). We conjecture that this might be attributed to the larger compounding error in BC. To shield the influence of compounding errors, we design an experiment where we measure the residual error of the next state after filling the historical contexts in the learned LanMDP context 4 and BC context 4 with expert states, rather than sampled states. The errors are both around 0.0004 for LanMDP and BC, closing the gap in Fig. 2f. The implication seems to be LanMDP is more robust to compounding errors than BC.

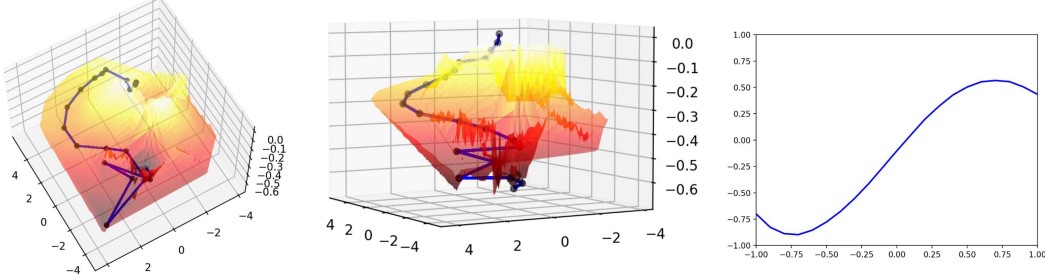

Figure 3: **Mapping a generated curve to a trajectory in the value landscape.** We train a neural network to approximate the non-Markovian value function constructed with the learned policy and transition following Theorem 1, and then visualize the landscape by projecting all history-augmented states to a 2D space with a top view (left) and a front view (middle). Starting from a random initial state, decisions are sequentially made according to the learned policy, leaving a curve in the original state space (right) and a trajectory on the value landscape. It is evident that the non-Markovian value increases monotonically along the trajectory.

To verify our analysis in Section 4, we visualize the non-Markovian value function defined in Theorem 1 in Fig. 3. The value increases monotonically when the policy generates the cubic curve step by step. In an animation we included on the project homepage[1], we further show that the action sampling at each state yields the highest value in reachable next states.

At last, we study repurposing the learned sequence model for goal-reaching. This is inspired by a surprising phenomenon, over-imitation, from psychology. Over-imitation occurs when imitators copy actions unnecessary for goal-reaching. In a seminal study [25], 3- to 4-year-old children and young chimpanzees were presented with a puzzle box containing a hidden treat. An experimenter demonstrated a goal-reaching sequence with both causally necessary and unnecessary actions. When the box was opaque, both chimpanzees and children tended to copy all actions. However, when a transparent box was used such that the causal mechanisms became apparent, chimpanzees omitted unnecessary actions, while human children imitated them. As shown in Fig. 2(h), planning with the learned non-Markovian value indeed leads to casually unnecessary states, consistent with the demonstrations. Planning with designed Markov rewards produces causally shortest paths.

## 5.2 Mujoco control tasks

We also report the empirical results of our model and baseline models on MuJoCo control tasks: Cartpole-v1, Reacher-v2, Swimmer-v3, Hopper-v2 and Walker2d-v2. We train an expert for each task using PPO [45]. They are then used to generate 10 trajectories for each task as demonstrations. Actions are deleted in the state-only setting.

**Setting** We conduct a comparative analysis of LanMDP against several established imitation learning baselines including BC [46], BCO [37], GAIL [35], GAIFO [7], and OPOLO [38]. Note that BC and GAIL have access to action labels, positioning them as the control group. The experimental group includes state-only methods such as LanMDP, BCO, GAIFO, and OPOLO. The expert is the idealized baseline. For all tasks, we adopt the MLP architecture for both transition and policy. The input and output dimensions are adapted to the state and action spaces in different tasks, and so are short-run sampling steps. Sequential contexts are extracted from stored episodic memory. The number of neurons in the input and hidden layer in the policy MLP varies according to the context length. We use replay buffers to store the self-interaction experiences for training the transition model offline. See Appendix D for detailed information on network architectures and hyper-parameters.

**Results** Results for context length 1 are illustrated through learning curves and a bar plot in Fig. 4. These learning curves are the average progress across 5 seeds. Scores in the bar plot are normalized relative to the expert score. Our model demonstrates significantly steeper learning curves compared to the state-only GAIFO baselines, especially in Cartpole and Walker2d. This illustrates the remarkable data efficiency of model-based methods. Additionally, LanMDP consistently matches or surpasses the performance of BC and GAIL, despite the latter having access to action labels. In comparison to the expert, LanMDP only lags behind in the most complex Walker2d task. However, it still maintains a noticeable margin over other state-only baselines.

---

[1]https://sites.google.com/view/non-markovian-decision-making

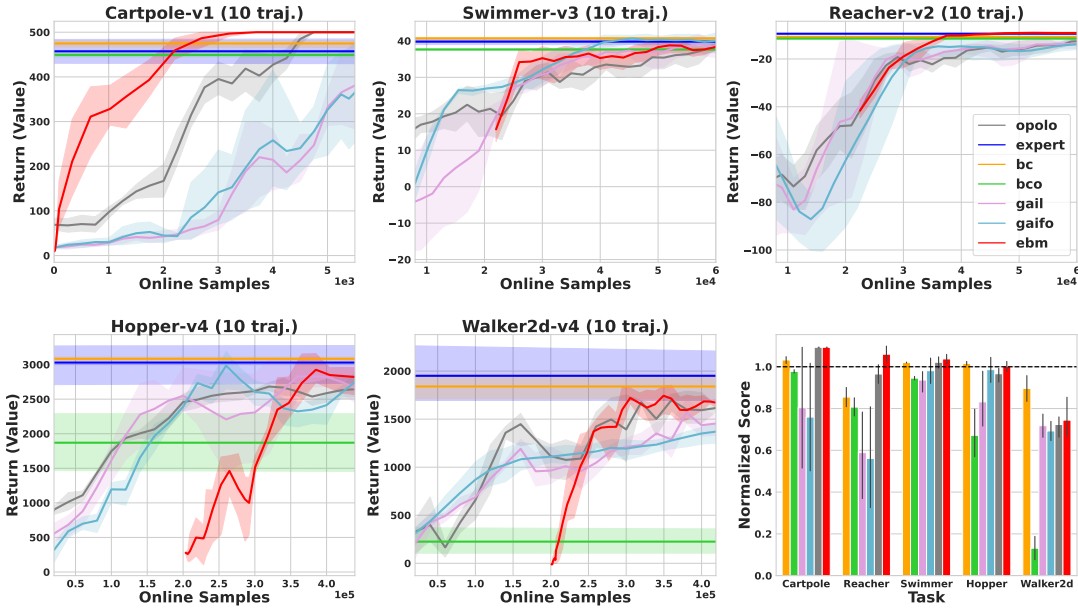

Figure 4: **Results in MuJoCo.** for our LanMDP (red), BC (orange), BCO (green), GAIL (purple), GAIFO (cyan), OPOLO (gray), expert (blue). The learning curves are obtained by averaging progress over 5 seeds. We only plot curves for interactive learning methods. The scores of all other methods are plotted as horizontal lines. LanMDP does not have performance scores in the first $K$ steps because this data is collected with random policy to fill the replay buffer, which is then used to train the transition model. $K = 0$ for Cartpole, $2e4$ for Reacher and Swimmer, $2e5$ for Hopper and Walker2d. We include these steps for fair comparisons. LanMDP outperforms existing state-only methods and matches BC, the best-performing state-action counterpart. The bar plot presents the scores from the best-performing policy during the training process, averaged across 5 seeds and normalized with the expert mean. The score in Reacher is offset by a constant before the division of the expert mean to align with the positive scores in all other tasks. The expert mean is plotted as a horizontal line. Our model clearly stands out in state-only methods, while matching and even outperforming those with action labels. Its scores only lag behind the expert mean in the most complex task. Better viewed in color.

Table 1: **Comparison between Markovian and non-Markovian policy in MuJoCo control task.** Context length is the number of prior sequential states that the policy depends on, with the current one included. Recall that these MuJoCo tasks are inherently Markovian, thanks to highly specified state features. Nevertheless, non-Markovian policies perform on par with Markovian ones and BC, despite having higher expressivity than sufficient. The best and the second-best results are highlighted. Results are averaged over 5 random seeds.

| TASK | CONTEXT 3 | CONTEXT 2 | CONTEXT 1 | BC |
|---|---|---|---|---|
| CARTPOLE | **500.00±0.00** | **500.00±0.00** | **500.00±0.00** | 474.80±18.87 |
| REACHER | -10.91±0.73 | -9.70±0.64 | -9.00±0.87 | **-8.76±0.12** |
| SWIMMER | **42.67±4.66** | **43.52±4.31** | 41.22±2.67 | 38.64±1.76 |
| HOPPER | 3051.16±111.78 | 3053.91±176.5 | 3045.27±240.45 | **3083.32±156.61** |
| WALKER2D | 1703.02±228.86 | **1811.77±369.54** | 1753.46±193.69 | **1839.94±376.87** |

Results for longer context lengths, *i.e.* the non-Markovian setting, are reported in Table 1, in which the highest return across the training process is listed. Originally invented for studying differentiable dynamics, MuJoCo offers state features that are inherently Markovian. Though a MDP is sufficiently expressive, learning a more generalized nMDP does not impair the performance. Sometimes it can even improve a little bit. Due to the limit of time, the maximum context length is only 3. Within the investigated regime, our result is consistent with that reported by Janner et al. [12]. We leave the experiments with longer memory and more sophisticated neural networks to future research.

Table 2 is a study of the computational overhead for the sampling techniques involved. The short-run MCMC for posterior inference takes longer than a single step of gradient descent. Replacing it with the proposed importance sampling improves training efficiency by a large margin.

Table 2: Computational overheads for posterior sampling, importance sampling, and gradient descent (in seconds) in one training step. MLP($n$,$m$) means that we implement the policy model as an MLP with $m$ layers and $n$ hidden neurons each layer. Results are averaged over 10 epochs. The number of MCMC steps is set to 10, 50 respectively. Replacing posterior sampling with importance sampling improves training efficiency.

| TASK | $a$ DIM | $s$ DIM | ARCHITECTURE | 10 STEPS | 50 STEPS |
|------|---------|---------|--------------|----------|----------|
| REACHER | 2 | 11 | MLP(150;4) | 0.0108/0.0076/0.0014 | 0.0480/0.0350/0.0014 |
| SWIMMER | 2 | 8 | MLP(150;4) | 0.0100/0.0071/0.0014 | 0.0463/0.0340/0.0014 |
| HOPPER | 3 | 12 | MLP(512;4) | 0.0268/0.0170/0.0074 | 0.1403/0.0836/0.0073 |
| WALKER2D | 6 | 18 | MLP(512;4) | 0.0282/0.0184/0.0077 | 0.1487/0.0899/0.0076 |

## 6 Discussion

**Related work in imitation learning** Earliest works in imitation learning utilized BC [3, 47]. When the training data is limited, temporal drifting in trajectories [48, 49] may occur, which led to the development of IRL [6, 50, 51, 4, 34, 35]. In recent years, the availability of abundant sequence/video data is not the primary concern, but rather the difficulty in obtaining action labels. There has since been increasing attention in ILfO [52, 53, 7, 38, 8], a setting similar to ours. Distinguished from existing ILfO solutions, our model probabilistically describes the entire trajectory. In particular, the energy-based model [54, 55] in the latent policy space [16] has been relatively unexplored. Additionally, the capability for model-based planning is also a novel contribution.

**Limitation and potential impact** The proposed model factorizes the joint distribution of state-action sequences into a time-invariant causal transition and a latent policy modulated by sequential contexts. While this model requires sampling methods, and can be non-negligible for higher-dimensional actions, it is worth noting that action quantization, as employed in transformer-based models [12, 13], has the potential to reduce the computation overhead. In our experiments, a measure of the diversity of behavior is omitted, similar to other works in the literature of reinforcement learning. However, it deserves further investigation since multi-modal density matching is a crucial metric in generative modeling. Importantly, our training objective and analysis are independent of specific modeling and sampling techniques, as long as the state transition remains time-invariant. Given the ability of neural networks to learn approximate invariance through data augmentation [56–59], we anticipate that our work will inspire novel training and inference techniques for monolithic sequential decision-making models [12–15].

**Implications in neuroscience and psychology** The proposed latent model is an amenable framework for studying the emergent patterns in the mirror neurons [60, 61], echoing recent studies in grid cells and place cells [62, 63]. When the latent action is interpreted as an internal intention, the inference process is a manifestation of Theory of Mind (ToM) [64]. The phenomenon of over-imitation [25, 26, 65] can also be relevant. As shown in Section 5, although the proposed model learns a causal transition and hence understands causality, when repurposed for goal-reaching tasks, the learned non-Markovian value can result in "unnecessary" state visitation. It would be interesting to explore if over-imitation is simply an overfitting due to excessive expressivity in sequence models.

## 7 Conclusion

In this study, we explore deep generative modeling of state-only sequences in non-Markovian domains. We propose a model, LanMDP, in which the policy functions as an energy-based prior within the latent space of the state transition generator. This model learns by EM-style maximum likelihood estimation. Additionally, we demonstrate the existence of a decision-making problem inherent in such probabilistic inference, providing a fresh perspective on maximum entropy reinforcement learning. To showcase the importance of non-Markovian dependency and evaluate the effectiveness of our proposed model, we introduce a specific experiment called cubic curve planning. Our empirical results also demonstrate the robust performance of LanMDP across the MuJoCo suite.

**Acknowledgment** AQ, QL and SZ are supported by the National Key R&D Program of China (2021ZD0150200). FG contributed to this work during his PhD study at UCLA. SX is supported by a Teaching Assistantship from UCLA CS. The authors thank Prof. Ying Nian Wu at UCLA, Baoxiong

Jia and Xiaojian Ma at BIGAI for their useful discussion. The authors would also like to thank the reviewers for their valuable feedback.

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

# A  Learning and Sampling

## A.1  Deep generative modelling

A complete trajectory is denoted by

$$\zeta = \{s_0, a_0, s_1, a_1, \cdots, a_{T-1}, s_T\}, \tag{1}$$

where $T$ is the maximum length of all observed trajectories. The joint distribution of state and action sequences can be factorized according to the causal assumptions in nMDP:

$$\begin{aligned}
p_\theta(\zeta) &= p(s_0) p_\alpha(a_0|s_0) p_\beta(s_1|s_0, a_0) \cdots p_\alpha(a_{T-1}|s_{0:T-1}) p_\beta(s_T|s_{T-1}, a_{T-1}) \\
&= p(s_0) \prod_{t=0}^{T-1} p_\alpha(a_t|s_{0:t}) p_\beta(s_{t+1}|s_t, a_t),
\end{aligned} \tag{2}$$

where $p_\alpha(a_t|s_{0:t-1})$ is the policy model with parameter $\alpha$, $p_\beta(s_t|s_{t-1}, a_{t-1})$ is the transition model with parameter $\beta$, both of which are parameterized with neural networks, $\theta = (\alpha, \beta)$. $p(s_0)$ is the initial state distribution, which can be sampled as a black box.

The density families of policy and transition are consistent with the conventional setting of IRL [4]. The transition describes the predictable change in state space, which is often possible to express the random variable $s_{t+1}$ as a deterministic variable $s_{t+1} = g_\beta(s_t, a_t, \epsilon)$, where $\epsilon$ is an auxiliary variable with independent marginal $p(\epsilon)$, and $g_\beta(.)$ is some vector-valued function parameterized by $\beta$. The policy accounts for bounded rationality as a Boltzmann distribution with state-action value as the unnormalized energy:

$$p_\alpha(a_t|s_{0:t}) = \frac{1}{Z(\alpha, s_{0:t})} \exp\left(f_\alpha(a_t; s_{0:t})\right), \tag{3}$$

where $f_\alpha(a_t; s_{0:t})$ is the negative energy, $Z(\alpha, s_{0:t}) = \int \exp(f_\alpha(a_t; s_{0:t})) da_t$ is the normalizing constant given the history $s_{0:t}$.

Since we can only observe state sequences, the aforementioned generative model can be understood as a sequential variant of LEBM [16], where the transition serves as the generator and the policy is a history-conditioned latent prior. The marginal distribution of state sequences and the posterior distribution of action sequences are:

$$p_\theta(s_{0:T}) = \int p_\theta(s_{0:T}, a_{0:T-1}) da_{0:T-1}, \quad p_\theta(a_{0:T-1}|s_{0:T}) = \frac{p_\theta(s_{0:T}, a_{0:T-1})}{p_\theta(s_{0:T})}. \tag{4}$$

## A.2  Maximum likelihood learning

We need to estimate $\theta = (\alpha, \beta)$. Suppose we observe training examples: $\{\xi^i\}, i = 1, 2, \cdots, n, \quad \xi^i = [s_0^i, s_1^i, ..., s_T^i]$. The log-likelihood function is:

$$L(\theta) = \sum_{i=1}^n \log p_\theta(\xi^i). \tag{5}$$

Denote posterior distribution of action sequence $p_\theta(a_{0:T-1}|s_{0:T})$ as $p_\theta(A|S)$ for convenience where $A$ and $S$ means the complete action and state sequences in a trajectory. The gradient of log-likelihood is:

$$\begin{aligned}
\nabla_\theta \log p_\theta(\xi) &= \nabla_\theta \log p_\theta(s_0, s_1, \cdots, s_T) \\
&= \mathbb{E}_{p_\theta(A|S)}[\nabla_\theta \log p_\theta(s_0, s_1, \cdots, s_T)] \\
&= \mathbb{E}_{p_\theta(A|S)}[\nabla_\theta \log p_\theta(s_0, s_1, \cdots, s_T)] + \mathbb{E}_{p_\theta(A|S)}[\nabla_\theta \log p_\theta(A|S)] \\
&= \mathbb{E}_{p_\theta(A|S)}[\nabla_\theta \log p_\theta(s_0, a_0, s_1, a_1, \cdots, a_{T-1}, s_T)] \\
&= \mathbb{E}_{p_\theta(A|S)}[\nabla_\theta \log p(s_0) p_\alpha(a_0|s_0) \cdots p_\alpha(a_{T-1}|s_{0:T-1}) p_\beta(s_T|s_{T-1}, a_{T-1})] \quad (6) \\
&= \mathbb{E}_{p_\theta(A|S)}[\nabla_\theta \sum_{t=0}^{T-1} (\log p_\alpha(a_t|s_{0:t}) + \log p_\beta(s_{t+1}|s_t, a_t))] \\
&= \mathbb{E}_{p_\theta(A|S)}[\sum_{t=0}^{T-1} (\underbrace{\nabla_\alpha \log p_\alpha(a_t|s_{0:t})}_{\text{policy/prior}} + \underbrace{\nabla_\beta \log p_\beta(s_{t+1}|s_t, a_t)}_{\text{transition}})],
\end{aligned}$$

where the third equation is because of a simple identity $\mathbb{E}_{\pi_\theta(a)}\left[\nabla_\theta \log \pi_\theta(a)\right] = 0$ for any probability distribution $\pi_\theta(a)$. Applying this simple identiy, we also have:

$$
\begin{aligned}
0 &= \mathbb{E}_{p_\alpha(a_t|s_{0:t})}\left[\nabla_\alpha \log p_\alpha(a_t|s_{0:t})\right] \\
&= \mathbb{E}_{p_\alpha(a_t|s_{0:t})}\left[\nabla_\alpha f_\alpha(a_t; s_{0:t}) - \nabla_\alpha \log Z(\alpha, s_{0:t})\right] \\
&= \mathbb{E}_{p_\alpha(a_t|s_{0:t})}\left[\nabla_\alpha f_\alpha(a_t; s_{0:t})\right] - \nabla_\alpha \log Z(\alpha, s_{0:t}).
\end{aligned}
\tag{7}
$$

Due to the normalizing constant $Z(\alpha, s_{0:t})$ in the energy-based prior $p_\alpha$, the gradient for the policy term involves both posterior and prior samples:

$$
\begin{aligned}
\delta_{\alpha,t}(S) &= \mathbb{E}_{p_\theta(A|S)}\left[\nabla_\alpha \log p_\alpha(a_t|s_{0:t})\right] \\
&= \mathbb{E}_{p_\theta(A|S)}\left[\nabla_\alpha f_\alpha(a_t; s_{0:t}) - \nabla_\alpha \log Z(\alpha, s_{0:t})\right] \\
&= \mathbb{E}_{p_\theta(A|S)}\left[\nabla_\alpha f_\alpha(a_t; s_{0:t}) - \mathbb{E}_{p_\alpha(a_t|s_{0:t})}\left[\nabla f_\alpha(a_t; s_{0:t})\right]\right] \\
&= \mathbb{E}_{p_\theta(A|S)}\left[\nabla_\alpha f_\alpha(a_t; s_{0:t})\right] - \mathbb{E}_{p_\alpha(a_t|s_{0:t})}\left[\nabla_\alpha f_\alpha(a_t; s_{0:t})\right],
\end{aligned}
\tag{8}
$$

where $\delta_{\alpha,t}(S)$ denotes the surrogate loss of policy term for time step $t$. Intuition can be gained from the perspective of adversarial training [34, 35]: On one hand, the model utilizes action samples from the posterior $p_\theta(A|S)$ as pseudo-labels to supervise the unnormalized prior at each step $p_\alpha(a_t|s_{0:t})$. On the other hand, it discourages action samples directly sampled from the prior. The model converges when prior samples and posterior samples are indistinguishable.

To ensure the transition model's validity, it needs to be grounded in real-world dynamics when jointly learned with the policy. Otherwise, the agent would be purely hallucinating based on the demonstrations. Throughout the training process, we allow the agent to periodically collect self-interaction data with $p_\alpha(a_t|s_{0:t})$ and mix transition data from two sources with weight $w_\beta$:

$$
\delta_{\beta,t}(S) = w_\beta \mathbb{E}_{p_\theta(A|S)}\left[\nabla_\beta \log p_\beta(s_{t+1}|s_t, a_t)\right] + (1-w_\beta)\mathbb{E}_{p_\alpha(a_t|s_{0:t}),Tr}\left[\nabla_\beta \log p_\beta(s_{t+1}|s_t, a_t)\right].
\tag{9}
$$

### A.3 General transition model

We need to compute the gradient of $\beta$ for the logarithm of transition probability in Equation 9, and as we will see in section 3.3, we also need to compute the gradient of the action during sampling actions. The reparameterization [43] is useful since it can be used to rewrite an expectation w.r.t $p_\beta(s_{t+1}|s_t, a_t)$ such that the Monte Carlo estimate of the expectation is differentiable, so we use delta function $\delta(.)$ to rewrite probability as an expectation:

$$
\begin{aligned}
p_\beta(s_{t+1}|s_t, a_t) &= \int \delta(s_{t+1} - s'_{t+1}) p_\beta(s'_{t+1}|s_t, a_t) ds'_{t+1} \\
&= \int \delta(s_{t+1} - g_\beta(s_t, a_t, \epsilon)) p(\epsilon) d\epsilon.
\end{aligned}
\tag{10}
$$

Taking advantage of the properties of $\delta(.)$:

$$
\int f(x)\delta(x)dx = f(0), \quad \delta(f(x)) = \Sigma_n \frac{1}{|f'(x_n)|}\delta(x - x_n),
\tag{11}
$$

where $f$ is differentiable and have isolated zeros, which is $x_n$, we can rewrite the transition probability as:

$$
\begin{aligned}
p_\beta(s_{t+1}|s_t, a_t) &= \int \sum_n \frac{1}{|\frac{\partial}{\partial\epsilon}g_\beta(s_t, a_t, \epsilon)|_{\epsilon=\epsilon_n}|}\delta(\epsilon - \epsilon_n)p(\epsilon)d\epsilon \\
&= \sum_n \frac{p(\epsilon_n)}{|\frac{\partial}{\partial\epsilon}g_\beta(s_t, a_t, \epsilon)|_{\epsilon=\epsilon_n}|},
\end{aligned}
\tag{12}
$$

where $\epsilon_n$ is the zero of $s_{t+1} = g_\beta(s_t, a_t, \epsilon)$. Therefore, if we have a differentiable simulator $\nabla_{a_t} \log p_\beta(s_{t+1}|s_t, a_t)$ and the analytical form of $p(\epsilon)$, then gradient of both $a_t$ and $\beta$ for $\log p_\beta(s_{t+1}|s_t, a_t)$ can be computed.

The simplest situation is:

$$
s_{t+1} = g_\beta(s_t, a_t) + \epsilon, \epsilon \sim p(\epsilon) = \mathcal{N}(0, \sigma^2).
\tag{13}
$$

In this case, there is only one zero $\epsilon^*$ for the transition function, $s_{t+1} = g_\beta(s_t, a_t) + \epsilon^*$, and the gradient of log probability is:

$$\begin{aligned}
\nabla \log p_\beta(s_{t+1}|s_t, a_t) &= \nabla \log \frac{p(\epsilon^*)}{|\frac{\partial}{\partial \epsilon}(g_\beta(s_t, a_t) + \epsilon)|_{\epsilon = \epsilon^*}|} \\
&= \nabla \log p(\epsilon^*) \\
&= \nabla \log p(s_{t+1} - g_\beta(s_t, a_t)) \\
&= \frac{1}{\sigma^2}(s_{t+1} - g_\beta(s_t, a_t))\nabla g_\beta(s_t, a_t).
\end{aligned} \tag{14}$$

### A.4 Prior and posterior sampling

The maximum likelihood estimation requires samples from the prior and the posterior distributions of actions. It would not be a problem if the action space is quantized. However, since we target general latent action learning, we proceed to introduce sampling techniques for continuous actions.

When sampling from a continuous energy space, short-run Langevin dynamics [17] can be an efficient choice. For a target distribution $\pi(a)$, Langevin dynamics iterates $a_{k+1} = a_k + s\nabla_{a_k} \log \pi(a_k) + \sqrt{2s}\epsilon_k$, where $k$ indexes the number of iteration, $s$ is a small step size, and $\epsilon_k$ is the Gaussian white noise. $\pi(a)$ can be either the prior $p_\alpha(a_t|s_{0:t})$ or the posterior $p_\theta(A|S)$. One property of Langevin dynamics that is particularly amenable for EBM is that we can get rid of the normalizing constant. So for each $t$ the iterative update for prior samples is

$$a_{t,k+1} = a_{t,k} + s\nabla_{a_{t,k}} f_\alpha(a_{t,k}; s_{0:t}) + \sqrt{2s}\epsilon_k. \tag{15}$$

Given a state sequence $s_{0:T}$ from the demonstrations, the posterior samples at each time step $a_t$ come from the conditional distribution $p(a_t|s_{0:T})$. Notice that with Markov transition, we can derive

$$p_\theta(a_{0:T-1}|s_{0:T}) = \prod_{t=0}^{T-1} p_\theta(a_t|s_{0:T}) = \prod_{t=0}^{T-1} p_\theta(a_t|s_{0:t+1}). \tag{16}$$

The point is, given the previous and the next subsequent state, the posterior can be sampled at each step independently. So the posterior iterative update is

$$\begin{aligned}
a_{t,k+1} &= a_{t,k} + s\nabla_{a_{t,k}} \log p_\theta(a_{t,k}|s_{0:t+1}) + \sqrt{2s}\epsilon_k \\
&= a_{t,k} + s\nabla_{a_{t,k}} \log p_\theta(s_{0:t}, a_{t,k}, s_{t+1}) + \sqrt{2s}\epsilon_k \\
&= a_{t,k} + s\nabla_{a_{t,k}} (\underbrace{\log p_\alpha(a_{t,k}|s_{0:t})}_{\text{policy/prior}} + \underbrace{\log p_\beta(s_{t+1}|s_t, a_t)}_{\text{transition}}) + \sqrt{2s}\epsilon_k.
\end{aligned} \tag{17}$$

Intuitively, action samples at each step are updated with the energy of all subsequent actions and a single-step forward by back-propagation. However, while gradients from the transition term are analogous to the inverse dynamics in BCO [37], it may lead to poor training performance due to non-injectiveness in forward dynamics [38].

We develop an alternative posterior sampling method with importance sampling to overcome this challenge. Leveraging the learned transition, we have

$$p_\theta(a_t|s_{0:t+1}) = \frac{p_\beta(s_{t+1}|s_t, a_t)}{\mathbb{E}_{p_\alpha(a_t|s_{0:t})}[p_\beta(s_{t+1}|s_t, a_t)]} p_\alpha(a_t|s_{0:t}). \tag{18}$$

Let $c(a_t; s_{0:t+1}) = \mathbb{E}_{p_\alpha(a_t|s_{0:t})}[p_\beta(s_{t+1}|s_t, a_t)]$, posterior sampling from $p_\theta(a_{0:T-1}|s_{0:T})$ can be realized by adjusting importance weights of independent samples from the prior $p_\alpha(a_t|s_{0:t})$, in which the estimation of weights involves another prior sampling. In this way, we avoid back-propagating through non-injective dynamics and save some computation overhead in Eq. (17).

To train the policy, Eq. (8) can now be rewritten as

$$\delta_{\alpha,t}(S) = \mathbb{E}_{p_\alpha(a_t|s_{0:t})}\left[\frac{p_\beta(s_{t+1}|s_t, a_t)}{c(a_t; s_{0:t+1})}\nabla_\alpha f_\alpha(a_t; s_{0:t})\right] - \mathbb{E}_{p_\theta(a_t|s_{0:t})}[\nabla_\alpha f_\alpha(a_t; s_{0:t})]. \tag{19}$$

### A.5 Algorithm

The learning and sampling algorithms with MCMC and with importance sampling for posterior sampling are described in Algorithm 1 and Algorithm 2.

**Algorithm 1:** LanMDP without importance sampling

---

**Input:** Learning iterations $N$, learning rate for energy-based policy $\eta_\alpha$, learning rate for transition model $\eta_\beta$, initial parameters $\theta_0 = (\alpha_0, \beta_0)$, expert demonstrations $\{s_{0:H}\}$, context length $L$, batch size $m$, number of prior and posterior sampling steps $\{K_0, K_1\}$, prior and posterior sampling step sizes $\{s_0, s_1\}$.
**Output:** $\theta_N = (\alpha_N, \beta_N)$.
Reorganize $\{s_{0:H}\}$ to to state sequenec segments $(s_{t-L+1}, \cdots, s_{t+1})$ with length $L + 1$.
Use energy-based policy with $\alpha_0$ collect transitions to fill in the replay buffer.
Use transitions in replay buffer to pre-train transition model $\beta_0$.
**for** $t = 0$ **to** $N - 1$ **do**
    **Demo sampling** Sample observed examples $(s_{t-L+1}, \cdots, s_{t+1})_{i=1}^m$.
    **Posterior sampling**: Sample $\{a_t\}_{i=1}^m$ using Eq. (17) with $K_1$ iterations and stepsize $s_1$.
    **Prior sampling**: Sample $\{\hat{a}_t\}_{i=1}^m$ using Eq. (15) with $K_0$ iterations and stepsize $s_0$.
    **Policy learning**: Update $\alpha_t$ to $\alpha_{t+1}$ by Eq. (8) with learning rate $\eta_\alpha$.
    **Transition learning**: Update replay buffer with trajectories from current policy model $\alpha_{t+1}$, then update $\beta_t$ to $\beta_{t+1}$ by Eq. (9) with learning rate $\eta_\beta$.
**end for**

---

**Algorithm 2:** LanMDP with importance sampling

---

**Input:** Learning iterations $N$, learning rate for energy-based policy $\eta_\alpha$, learning rate for transition model $\eta_\beta$, initial parameters $\theta_0 = (\alpha_0, \beta_0)$, expert demonstrations $\{s_{0:H}\}$, context length $L$, batch size $m$, number of prior sampling steps $K$ and step sizes $s$.
**Output:** $\theta_N = (\alpha_N, \beta_N)$.
Reorganize $\{s_{0:H}\}$ to to state sequenec segments $(s_{t-L+1}, \cdots, s_{t+1})$ with length $L + 1$.
Use energy-based policy with $\alpha_0$ collect transitions to fill in the replay buffer.
Use transitions in replay buffer to pre-train transition model $\beta_0$.
**for** $t = 0$ **to** $N - 1$ **do**
    **Demo sampling** Sample observed examples $(s_{t-L+1}, \cdots, s_{t+1})_{i=1}^m$.
    **Prior sampling**: Sample $\{\hat{a}_t\}_{i=1}^m$ using Eq. (15) with $K_0$ iterations and stepsize $s_0$.
    **Policy learning**: Update $\alpha_t$ to $\alpha_{t+1}$ by Eq. (19) with learning rate $\eta_\alpha$.
    **Transition learning**: Update replay buffer with trajectories from current policy model $\alpha_{t+1}$, then update $\beta_t$ to $\beta_{t+1}$ by Eq. (9) with learning rate $\eta_\beta$.
**end for**

---

# B A Decision-making Problem in MLE

Let the ground-truth distribution of demonstrations be $p^*(s_{0:T})$, and the learned marginal distributions of state sequences be $p_\theta(s_{0:T})$. Eq. (5) is an empirical estimate of

$$\mathbb{E}_{p*(s_{0:T})}[\log p_\theta(s_{0:T})] = \mathbb{E}_{p*(s_0)}\left[\log p^*(s_0) + \mathbb{E}_{p*(s_{1:T}|s_0)}[\log p_\theta(s_{1:T}|s_0)]\right]. \quad (20)$$

We can show that a sequential decision-making problem can be constructed to maximize the same objective. To start off, suppose the MLE yields the maximum, we will have $p_{\theta*} = p^*$.

Define $V^*(s_0) := \mathbb{E}_{p*(s_{1:T}|s_0)}[\log p^*(s_{1:T}|s_0)]$, we can generalize it to have a *V function*

$$V^*(s_{0:t}) := \mathbb{E}_{p*(s_{t+1:T}|s_{0:t})}[\log p^*(s_{t+1:T}|s_{0:t})], \quad (21)$$

which comes with a Bellman optimality equation:

$$V^*(s_{0:t}) = \mathbb{E}_{p*(s_{t+1}|s_{0:t})}\left[r(s_{t+1}, s_{0:t}) + V^*(s_{0:t+1})\right], \quad (22)$$

with $r(s_{t+1}, s_{0:t}) := \log p^*(s_{t+1}|s_{0:t}) = \log \int p_{\alpha*}(a_t|s_{0:t})p_{\beta*}(s_{t+1}|s_t, a_t)da_t$, $V^*(s_{0:T}) := 0$. It is worth noting that the $r$ defined above involves the optimal policy, which may not be known a priori. We can resolve this by replacing it with $r_\alpha$ for an arbitrary policy $p_\alpha(a_t|s_{0:t})$. All Bellman identities and updates should still hold. Anyways, involving the current policy in the reward function should not appear to be too odd given the popularity of maximum entropy RL [20, 24].

The entailed Bellman update, *value iteration*, for arbitrary $V$ and $\alpha$ is

$$V(s_{0:t}) = \mathbb{E}_{p*(s_{t+1}|s_{0:t})}\left[r_\alpha(s_{0:t}, s_{t+1}) + V(s_{0:t+1})\right]. \quad (23)$$

We then define $r(s_{t+1}, a_t, s_{0:t}) := r(s_{t+1}, s_{0:t}) + \log p_{\alpha*}(a_t|s_{0:t})$ to construct a $Q$ function:

$$Q^*(a_t; s_{0:t}) := \mathbb{E}_{p*(s_{t+1}|s_{0:t})}\left[r(s_{t+1}, a_t, s_{0:t}) + V^*(s_{0:t+1})\right], \quad (24)$$

which entails a Bellman update, *Q backup*, for arbitrary $\alpha$, $Q$ and $V$

$$Q(a_t; s_{0:t}) = \mathbb{E}_{p*(s_{t+1}|s_{0:t})}\left[r_\alpha(s_{0:t}, a_t, s_{t+1}) + V(s_{0:t+1})\right]. \quad (25)$$

Also note that the $V$ and $Q$ in identities Eq. (23) and Eq. (25) respectively are not necessarily associated with the policy $p_\alpha(a_t|s_{0:t})$. Slightly overloading the notations, we use $Q^\alpha$, $V^\alpha$ to denote the expected returns from policy $p_\alpha(a_t|s_{0:t})$.

By now, we finish the construction of atomic algebraic components and move on to check if the relations between them align with the algebraic structure of a sequential decision-making problem [9].

We first prove the construction above is valid at optimality.

**Lemma 1.** *When $f_\alpha(a_t; s_{0:t}) = Q^*(a_t; s_{0:t}) - V^*(s_{0:t})$, $p_\alpha(a_t|s_{0:t})$ is the optimal policy.*
*Proof.* Note that the construction gives us

$$\begin{aligned}
Q^*(a_t; s_{0:t}) &= \mathbb{E}_{p*(s_{t+1}|s_{0:t})}\left[r(s_{t+1}, s_{0:t}) + \log p_{\alpha*}(a_t|s_{0:t}) + V^*(s_{0:t+1})\right] \\
&= \log p_{\alpha*}(a_t|s_{0:t}) + \mathbb{E}_{p*(s_{t+1}|s_{0:t})}\left[r(s_{t+1}, s_{0:t}) + V^*(s_{0:t+1})\right] \\
&= \log p_{\alpha*}(a_t|s_{0:t}) + V^*(s_{0:t}).
\end{aligned} \quad (26)$$

Obviously, $Q^*(a_t; s_{0:t})$ lies in the hypothesis space of $f_\alpha(a_t; s_{0:t})$. $\square$

Lemma 1 indicates that we need to either parametrize $f_\alpha(a_t; s_{0:t})$ or $Q(a_t; s_{0:t})$.

While $Q^\alpha$ and $V^\alpha$ are constructed from the optimality, the derived $Q^\alpha$ and $V^\alpha$ measure the performance of an interactive agent when it executes with the policy $p_\alpha(a_t|s_{0:t})$. They should be consistent with each other.

**Lemma 2.** *$V^\alpha(s_{0:t})$ and $\mathbb{E}_{p_\alpha(a_t|s_{0:t})}[Q^\alpha(a_t; s_{0:t})]$ yield the same optimal policy $p_{\alpha*}(a_t|s_{0:t})$.*
*Proof.*

$$\begin{aligned}
&\mathbb{E}_{p_\alpha(a_t|s_{0:t})}[Q^\alpha(a_t; s_{0:t})] := \mathbb{E}_{p_\alpha(a_t|s_{0:t})}\left[\mathbb{E}_{p*(s_{t+1}|s_{0:t})}\left[r(s_{t+1}, a_t, s_{0:t}) + V^\alpha(s_{0:t+1})\right]\right] \\
&= \mathbb{E}_{p_\alpha(a_t|s_{0:t})}\left[\mathbb{E}_{p*(s_{t+1}|s_{0:t})}\left[\log p_\alpha(a_t|s_{0:t}) + r(s_{t+1}, s_{0:t}) + V^\alpha(s_{0:t+1})\right]\right] \\
&= \mathbb{E}_{p*(s_{t+1}|s_{0:t})}\left[r(s_{t+1}, s_{0:t}) - \mathcal{H}_\alpha(a_t|s_{0:t}) + V^\alpha(s_{0:t+1})\right] \\
&= V^\alpha(s_{0:t}) - \mathcal{H}_\alpha(a_t|s_{0:t}) - \sum_{k=t+1}^{T-1}\mathbb{E}_{p*(s_{t+1:k}|s_{0:t})}[\mathcal{H}_\alpha(a_k|s_{0:k})],
\end{aligned} \quad (27)$$

where the last line is derived by recursively applying the Bellman equation in the line above until $s_{0:T}$ and then applying backup with Eq. (23). As an energy-based policy, $p_\alpha(a_t|s_{0:t})$'s entropy is inherently maximized [66]. Therefore, within the hypothesis space, $p_{\alpha*}(a_t|s_{0:t})$ that optimizes $V^\alpha(s_{0:t})$ also leads to optimal expected return $\mathbb{E}_{p_\alpha(a_t|s_{0:t})}[Q^\alpha(a_t; s_{0:t})]$. $\square$

If we parametrize the policy as $p_\alpha(a_t|s_{0:t}) \propto \exp(Q^\alpha(a_t; s_{0:t}))$, the logarithmic normalizing constant $\log Z^{\alpha_k}(s_{0:t})$ will be the *soft V function* in maximum entropy RL [21–23]

$$V^\alpha_{soft}(s_{0:t}) := \log \int \exp(Q^\alpha(a_t; s_{0:t}))da_t, \tag{28}$$

even if the reward function is defined differently. We can further show that Bellman identities and backup updates above can entail RL algorithms that achieve optimality of the decision-making objective $V^\alpha$, including *soft policy iteration* [20]

$$p_{\alpha_{k+1}}(a_t|s_{0:t}) \leftarrow \frac{\exp(Q^{\alpha_k}(a_t; s_{0:t}))}{Z^{\alpha_k}(s_{0:t})}, \forall s_{0:t}, k \in [0, 1, ...M]; \tag{29}$$

and *soft Q iteration* [21]

$$Q^{\alpha_{k+1}}(a_t; s_{0:t}) \leftarrow \mathbb{E}_{p*(s_{t+1}|s_{0:t})}\left[r_\alpha(s_{0:t}, a_t, s_{t+1}) + V^{\alpha_k}_{soft}(s_{0:t+1})\right], \forall s_{0:t}, a_t,$$
$$V^{\alpha_{k+1}}_{soft}(s_{0:t}) \leftarrow \log \int \exp(Q^{\alpha_k}(a; s_{0:t}))da, \forall s_{0:t}, k \in [0, 1, ...M]. \tag{30}$$

**Lemma 3.** *If $p*(s_{t+1}|s_{0:t})$ is accessible and $p_{\beta*}(s_{t+1}|s_t, a_t)$ is known, soft policy iteration and soft Q learning both converge to $p_{\alpha*}(a_t|s_{0:t}) = p_{Q*}(a_t|s_{0:t}) \propto \exp(Q^*(a_t; s_{0:t}))$ under certain conditions.*

*Proof.* See the convergence proof by Ziebart [20] for *soft policy iteration* and the proof by Fox et al. [21] for *soft Q learning*. The latter requires Markovian assumption. But under some conditions, it can be extended to non-Markovian domains in the same way as proposed by Majeed and Hutter [67]. $\square$

Lemma 3 means given $p*(s_{t+1}|s_{0:t})$ and $p_{\beta*}(s_{t+1}|s_t, a_t)$, we can recover $p_{\alpha*}$ through reinforcement learning methods, instead of the proposed MLE. So $p_\alpha(a_t|s_{0:t})$ is a viable policy space for the constructed sequential decision-making problem.

Together, Lemma 1, Lemma 2 and Lemma 3 provide constructive proof for a valid sequential decision-making problem that maximizes the same objective of MLE, described by Theorem 1.

**Theorem 1.** *Assuming the Markovian transition $p_{\beta*}(s_{t+1}|s_t, a_t)$ is known, the ground-truth conditional state distribution $p*(s_{t+1}|s_{0:t})$ for demonstration sequences is accessible, we can construct a sequential decision-making problem, based on a reward function $r_\alpha(s_{t+1}, s_{0:t}) := \log \int p_\alpha(a_t|s_{0:t})p_{\beta*}(s_{t+1}|s_t, a_t)da_t$ for an arbitrary energy-based policy $p_\alpha(a_t|s_{0:t})$. Its objective is*

$$\sum_{t=0}^{T} \mathbb{E}_{p*(s_{0:t})}[V^{p_\alpha}(s_{0:t})] = \mathbb{E}_{p*(s_{0:T})}\left[\sum_{t=0}^{T}\sum_{k=t}^{T} r_\alpha(s_{k+1}; s_{0:k})\right],$$

*where $V^{p_\alpha}(s_{0:t}) := E_{p*(s_{t+1:T}|s_{0:t})}[\sum_{k=t}^{T} r_\alpha(s_{k+1}; s_{0:k})]$ is the value function for $p_\alpha$. This objective yields the same optimal policy as the Maximum Likelihood Estimation $\mathbb{E}_{p*(s_{0:T})}[\log p_\theta(s_{0:T})]$.*

*If we further define a reward function $r_\alpha(s_{t+1}, a_t, s_{0:t}) := r_\alpha(s_{t+1}, s_{0:t}) + \log p_\alpha(a_t|s_{0:t})$ to construct a Q function for $p_\alpha$*

$$Q^{p_\alpha}(a_t; s_{0:t}) := \mathbb{E}_{p*(s_{t+1}|s_{0:t})}\left[r_\alpha(s_{t+1}, a_t, s_{0:t}) + V^{p_\alpha}(s_{0:t+1})\right].$$

*The expected return of $Q^{p_\alpha}(a_t; s_{0:t})$ forms an alternative objective*

$$\mathbb{E}_{p_\alpha(a_t|s_{0:t})}[Q^{p_\alpha}(a_t; s_{0:t})] = V^{p_\alpha}(s_{0:t}) - \mathcal{H}_\alpha(a_t|s_{0:t}) - \sum_{k=t+1}^{T-1} \mathbb{E}_{p*(s_{t+1:k}|s_{0:t})}[\mathcal{H}_\alpha(a_k|s_{0:k})]$$

*that yields the same optimal policy, for which the optimal $Q^*(a_t; s_{0:t})$ can be the energy function.*

*Only under certain conditions, this sequential decision-making problem is solvable through non-Markovian extensions of the maximum entropy reinforcement learning algorithms.*

## C   More results on Curve Planning

The energy function is parameterized by a small MLP with one hidden layer and $4 * L$ hidden neurons, where $L$ is the context length. In short-run Langevin dynamics, the number of samples, the number of sampling steps, and the stepsize are 4, 20 and 1 respectively. We use Adam optimizer with a learning rate 1e-4 and batch size 64. Here we present the complete result in Fig. A1 with different training steps under context length 1 2 4 6, the acceptance rate and residual error of the testing trajectories, as well as the behavior cloning results. We can see that even with sufficient context, BC performs worse than LanMDP. Also, from the result of context length 6 we can see that excessive expressivity does not impair performance, it just requires more training.

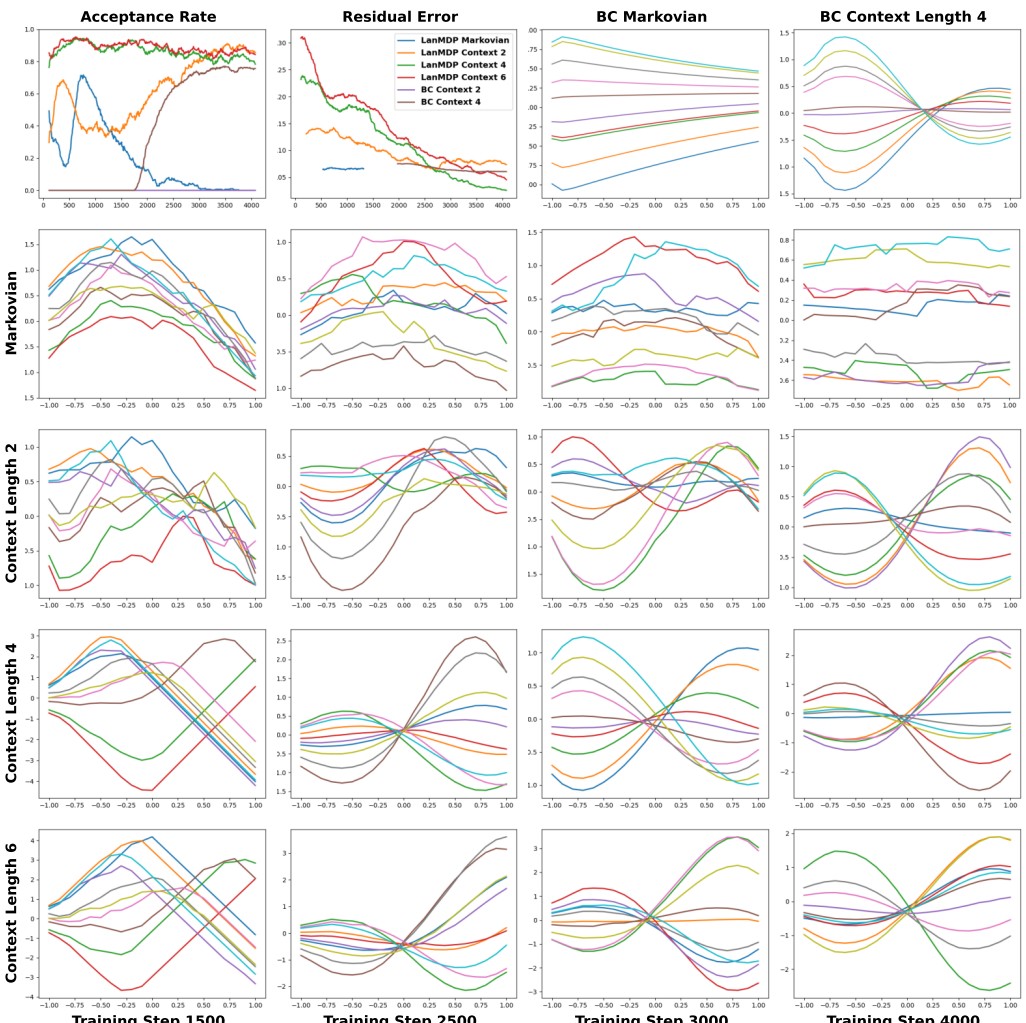

Figure A1: More results for cubic curve generation

## D   Implementation Details of MuJoCo Environment

This section delineates the configurations for the MuJoCo environments utilized in our research. In particular, we employ standard environment horizons of 500 and 50 for Cartpole-v1 and Reacher-v2, respectively. Meanwhile, for Swimmer-v2, Hopper-v2, and Walker2d-v2, we operate within an environment horizon set at 400 as referenced in previous literature [52, 68–72]. Additional specifications are made for Hopper-v2 and Walker2d-v2, where the velocity of the center of mass was integrated into the state parameterization [52, 68, 70, 72]. We leverage PPO [45] approach to train the expert policy until it reaches (approximately) 450, -10, 40, 3000, 2000 for Cartpole-v1, Reacher-v2,

Swimmer-v2, Hopper-v2, Walker2d-v2 respectively. It should be noted that all results disclosed in the experimental section represent averages over five random seeds. Comparative benchmarks include BC [46], BCO [37], GAIL [35], and GAIFO [7]. MobILE [52] is a recent method for Markovian model-based imitation from observation. However, we failed to reproduce the expected performance utilizing various sets of demonstrations, so it is prudently omitted from the present displayed result. We specifically point out that BC/GAIL algorithms are privy to expert actions, however, our algorithm is not. We report the mean of the best performance achieved by BC/BCO with five random seeds, even though these peak performances may transpire at varying epochs. For BC, we executed the supervised learning algorithm for 200 iterations. The BCO/GAIL algorithms are run with an equivalent number of online samples as LanMDP for a fair comparison. All benchmarking is performed using a single 3090Ti GPU and implemented using the PyTorch framework. Notably, in our codebase, the modified environments of Hopper-v2 and Walker2d-v2 utilize MobILE's implementation [52]. Referring to the results in the main text, our presentation of normalized results in bar graph form is derived by normalizing each algorithm's performance (mean/standard deviation) against the expert mean. For Reacher-v2, due to the inherently negative rewards, we first add a constant offset of 20 to each algorithm's performance, thus converting all values to positive before normalizing them against the mean of expert policy.

We parameterize both the policy model and the transition model as MLPs, and the non-linear activation function is Swish and LeakyReLU respectively. We use AdamW to optimize both policy and transition. To stabilize training, we prefer using actions around which the transition model is more certain for computing the expectation over importance-weighted prior distribution in Eq. (19). Therefore, we use a model ensemble with two transition models and use the disagreement between these two models to measure the uncertainty of the sampled actions. We implement Algorithm 2 for all experiments to avoid expensive computation of the gradient for the transition model in posterior sampling. As for better and more effective short-run Langevin sampling, we use a polynomially decaying schedule for the step size as recommended in [73]. We also use weakly L2 regularized energy magnitudes and clip gradient steps like [74], choosing to clip the total amount of change value, i.e. after the gradient and noise have been combined. To realize more delicate decision-making, another trick in Implicit Behavior Clone [75] is also adopted for the inference/testing stage that we continue running the MCMC chain after the step size reaches the smallest in the polynomial schedule until we get twice as many inference Langevin steps as were used during training.

Hyper-parameters are listed in Table 3. Other hyperparameters that are not mentioned here are left as default in PyTorch. Also, note that the Cartpole-v1 task has no parameters for sampling because expectation can be calculated analytically.

Table 3: Hyper-parameter list of MuJoCo experiments

| Parameter | Cartpole-v1 | Reacher-v2 | Swimmer-v2 | Hopper-v2 | Walker2d-v2 |
|---|---|---|---|---|---|
| **Environment Specification** | | | | | |
| Horizon | 500 | 50 | 400 | 400 | 400 |
| Expert Performance ($\approx$) | 450 | -10 | 40 | 3000 | 2000 |
| **Transition Model** | | | | | |
| Architecture(hidden;layers) | MLP(64;4) | MLP(64;4) | MLP(128;4) | MLP(512;4) | MLP(512;4) |
| Optimizer(LR) | 3e-3 | 3e-3 | 3e-3 | 3e-3 | 3e-3 |
| Batch Size | 2500 | 20000 | 20000 | 32768 | 32768 |
| Replay Buffer Size | 2500 | 20000 | 20000 | 200000 | 200000 |
| **Policy Model (with context length $L$)** | | | | | |
| Architecture(hidden;layers) | MLP(150*$L$;4) | MLP(150*$L$;4) | MLP(150*$L$;4) | MLP(512*$L$;4) | MLP(512*$L$;4) |
| Learning rate | 1e-3 | 1e-2 | 1e-2 | 1e-2 | 5e-3 |
| Batch Size | 2500 | 20000 | 20000 | 32768 | 32768 |
| Number of test trajectories | 5 | 20 | 20 | 50 | 50 |
| **Sampling Parameters** | | | | | |
| Number of prior samples | \ | 8 | 8 | 8 | 8 |
| Number of Langevin steps | \ | 100 | 100 | 100 | 100 |
| Langevin initial stepsize | \ | 10 | 10 | 10 | 10 |
| Langevin ending stepsize | \ | 1 | 1 | 1 | 1 |

