# OpenReview forum: "Learning non-Markovian Decision-Making from State-only Sequences"
_NeurIPS.cc/2023/Conference — NeurIPS 2023 poster_

### Official Review · Reviewer_wvit · 2023-07-02

**Soundness:** 4 excellent
**Presentation:** 3 good
**Contribution:** 3 good
**Rating:** 7
**Confidence:** 4

**Summary:**

This paper proposes an offline model-based method for learning a non-Markovian Decision Process (nMDP), i.e. the environment is Markovian but the policy is related to previous histories. In the dataset, we can only observe the sequence of states. This paper proposes a Lan-MDP algorithm, which estimates the transition and policy through MLE. Specifically, Lan-MDP leverages Langevin dynamics to calculate the weights in the posterior distribution. The algorithm then transforms behavior cloning into a reward-maximization problem by constructing a reward with the estimated transition/policy to do planning for sequential decisions.

**Strengths:**

The paper aims to tackle the issue of non-Markovian imitation learning where we can only observe a dataset of sequential state (with no actions), which is known to be very hard due to long history dependency (generally NP-hard, see [1]).

[1] Learning in Observable POMDPs, without Computationally Intractable Oracles, Noah Golowich, Ankur Moitra, Dhruv Rohatgi

**Weaknesses:**

In my humble opinion, I think the author could illustrate more examples (either in the real-world or synthetic problems) that satisfy the setting they proposed. POMDP should be one important example that is confounded/has a non-Markovian structure, but its transition doesn't have a Markovian structure. Intuitively, the history-dependency of the transition would cause difficulty in planning/learning in non-Markovian problems ([1]), and I am not sure Lan-MDP could tackle this.


[1] Learning in Observable POMDPs, without Computationally Intractable Oracles, Noah Golowich, Ankur Moitra, Dhruv Rohatgi

**Questions:**

Line 226: Additional 'including' could be a typo

**Limitations:**

See Weakness

---

> ### Author Rebuttal · Authors · 2023-08-10
>
> Thank you very much for your feedback!
>
>
> > Q: Could LanMDP tackle POMDP?
>
> Indeed, with the Markovian assumption in transition, the proposed model cannot directly extend to POMDP, where the transition is non-Markovian. However, it is worth a trial with the general modeling of latent EBM with non-Markovian transition and policy.

---

### Official Review · Reviewer_W5w2 · 2023-07-06

**Soundness:** 3 good
**Presentation:** 3 good
**Contribution:** 2 fair
**Rating:** 6
**Confidence:** 5

**Summary:**

The paper considers the problem of imitation learning from state observations in a setting where the transition dynamics are Markovian, but the (unknown) reward function and, therefore, the optimal policy are not. The problem is framed as maximizing the likelihood of the expert's state marginal based on a generative process that is parameterized via a history-based EBM-policy and a nonlinear Gaussian transition model (hence, the method is model-based). To optimize the likelihood of the state-only trajectory with respect to these action-related models, the method exploits that the gradient of the state-likelihood is equivalent to the expected gradient of the state-action likelihood, $\nabla\_{\theta}\log p\_{\theta}(\tau\_{S}) =  \nabla\_{\theta} E\_{p\_{\theta}(A|S)}[\log p\_{\theta}(\tau\_{S,A})]$, where $p(A|S)$ is the posterior distribution (which takes into account that $s\_{t+1}$ contains information about $a\_{t}$). As the optimization involves expectations with respect to intractable models (not only the posterior, but also the prior, that is, the EBM policy is intractable), the proposed method uses a combination of Langevian MCMC and importance sampling.

The method is tested on a toy-problem, where a 2D point-mass should get velocity-controlled towards a goal location, while following a 3rd-order polynomial path (which corresponds to a non-Markovian reward). When providing a sufficiently large history to the policy, it can indeed learn to exhibit the desired behavior.

In the main experiments on (Markovian) Mujoco locomotion tasks, the method is compared to GAIL, and behavioral cloning (BC), which both get groundtruth access to the expert actions, and their counterparts GAIfO, and behavioral cloning from observations (BCO) that only use state observations. In these experiments the proposed method outperforms the competing imitation learning from observation methods and matches or even outperforms the baselines that have priviledged information about expert actions.

**Strengths:**

- The proposed method is novel and the formulation as an MLE problem is interesting.

- The problem setting is very relevant. Imitation learning from observations is an important topic because oftentimes we can not observe the expert's actions. Also learning from non-Markovian policies is important, because for general tasks, we can not assume that the provided state-action space is Markovian.

- The claims seem correct. I checked the derivations and could not spot any issues.

- The paper is well-written and clear.

**Weaknesses:**

Technical Soundness
--------------------------

In the derivation that I sketched in the summary, the actions would not be grounded. The maximum likelihood objective would be an offline objective (similar to behavioral cloning), where the "actions" are latent variables with no particular meaning. Hence, for computing the gradient with respect to the transition model, the paper mixes in samples from policy rollouts. While grounding the latent variables in such manner is intuitively reasonable and based on the experiments sufficient to learn meaningful policies, it also seems to break the derivations. For all I can tell, the resulting method does no longer maximize the log-likelihood of the state trajectory.


Experiments
----------------

- In the toy experiments, behavioral cloning fails to produce the desired cubic trajectories, even when using a history-based policy. The paper states that this might be caused due to the fact that their BC implementation uses a Gaussian (and, thus, unimodal) policy, whereas the proposed method uses a much more expressive EBM. I don't see why the task would require a unimodal policy because with a sufficiently large history (>= 4), there is at most a single action that stays on a 3-rd order polynomial path. Furthermore, it would be straightforward to use a multimodal policy such as a normalizing flow, or for an even fairer comparison an EBM (which could also be trained using Langevian MC estimates of the partition function, or using more advanced techniques). Of course, the comparison to BC is not fair anyways, since it does not make use of online samples. I wonder why GAIfO was not trained in this toy setting.

- I could expect the method to outperform BC and BCO mainly due to the fact that those methods can suffer from compounding errors (BCO is online, but uses the interactions only for labeling expert actions using a learned inverse dynamic model). GAIL and GaifO are online methods, but they are not very sample efficient as they are not model-based, and so it's also not surprising that they underperform in the low data regime. Model-based imitation learning from observations alone (MobILE) would seem to be a more appropriate baselines, and as mentioned in the supplementary the modified Hopper and Walker environments where even taken from this work. Furthermore, least-squares inverse Q-Learning (LS-IQ) (Al-Hafez et al., 200) was tested in the imitation learning from observation setting using an inverse dynamic model, where it also cleraly outperformed GaifO.

- The method relies on MCMC during training, but also during inference. The submission does not evaluate the compuational overhead resulting from this design choice.

- Table 1 seems to be based on a single seed.

References
---------------
Al-Hafez, F., Tateo, D., Arenz, O., Zhao, G., & Peters, J. (2022). LS-IQ: Implicit Reward Regularization for Inverse Reinforcement Learning. In The Eleventh International Conference on Learning Representations.

**Questions:**

- It is not clear how the goal position is specified for the toy experiment. Is it added to the state space?

- Line 77 argues that it non-Markovian rewards and Markovian states are usually sufficient because we can always define a state space that makes the dynamics Markovian. Couldn't we just as well argue that we can always define a state space that makes the reward function Markovian?

- What is the computational overhead caused by MCMC?

- What is the motivation for using an EBM instead of a more tractable policy such as a normalizing flow?

- Line 192 states that Gaussian expectations can be approximated by the mean and refers to Kingma & Welling. What statement are you exactly referring to?

- Is there any theoretical justification for mixing in online samples in the maximum likelihood objective?

- Line 222. What is y' ? is it the target y location? Should it then also be $x' = 1$ instead of $x = 1$?

- What is the main insight of Section 4? That there exists a reward function such that we can frame the MLE problem as a MaxEnt-RL problem? What are the implications?

- Why is it overimitation if the agent imitates the cubic path of the expert? In the cited psychology experiments there was an mechanism and an extrinsic reward and therefore overimitation can be defined as imitating features that are not relevant for obtaining the extrinsic reward. How is overimitation defined in a setting without extrinsic reward?

**Limitations:**

While the paper mentions the limitation that MCMC may become prohibitively expensive, the paper does not provide evaluations of the computational cost. Also, MCMC can suffer from high variance and bad mixing time.

---

> ### Author Rebuttal · Authors · 2023-08-10
>
> Thank you for the feedback!
>
> > Learning objective
>
> The mixing of gradients in Eq.8  is derived from jointly optimizing two log-likelihood, Eq. 5 and $L_{online}(\beta) = \sum_{i=1}^{m} \sum_{t=1}^{T} \log p_\beta (s_{t+1}^i | s_{t}, s_t)$. We will include this explanation in the revision.
>
> > Confusion about BC's result with sufficient history in the toy experiment
>
> “Behavioral cloning fails to produce the desired cubic trajectories, even when using a history-based policy” is a misunderstanding. Thanks to this comment, we realize that putting the result of behavior cloning with sufficient history in the supplementary and leaving only the one with insufficient history in the main text could be confusing. Please kindly refer to the two subfigures at the bottom right corner of Fig. 1 in the supplementary for the result aligning with your prediction.
>
> Since there seems to be a typo in “I don't see why the task would require a unimodal policy”, we reckon that the reviewer may also be wondering about the gap between LanMDP context 4 and BC context 4 in Fig. 2f. We hypothesize that this might be attributed to the larger compounding error in BC. To isolate the influence of compounding error, we design an experiment where we measure the residual error of the next state after filling the historical contexts in the learned LanMDP context 4 and BC context 4 with expert states, rather than sampled states. Interestingly, the error is around 0.0004 for both LanMDP and BC, closing the gap in Fig. 2f. The implication seems to be LanMDP is more robust to compounding error than BC.
>
> > In the toy task, how about baseline models other than BC
>
> 1. Fair comparison with BC: As stated in L230-232, “in a deterministic environment, there should not be a difference between BC and BCO, as the latter basically employs inverse dynamics to recover action labels. For our model, this simple transition can either be learned or implanted. Empirically, we don’t notice a significant difference.”
>
> 2. Other multimodal latent policy such as normalizing flow model: We agree that it would be interesting to explore these models in future work. Our major consideration in choosing latent EBM is that it makes the least statistical assumptions (though the computational assumption of MCMC may be strong), such that the exposition of the decision-making process (Sec. 2) is clean to people from both literatures of generative modeling and decision making.
>
> 3. Comparison with EBM with action labels: In a deterministic environment with such a simple Markovian transition, sampling the posterior $p(a_t | s_{0: t+1})$ can be done by inverse dynamics $g(a_t | s_t, s_{t+1})$ in which $a_t$ can be uniquely identified. So there is actually no difference between latent EBM with grount-truth transition and EBM with action labels.
>
> 4. Comparison with GAIfO: Unlike BC which can be extended to non-Markovian setting, GAIFO, as a state-matching method, is limited to Markovian tasks.
>
> > Model-based baselines
>
> We have been experimenting with the public code of MobILE on our demonstration sets since we were preparing this submission but haven’t obtained results comparable to those reported in their paper. Unfortunately, the contacts we sent out to the authors have all gone unanswered.
>
> Thanks for pointing us to the LS-IQ paper. Since the major contribution of our work is not model-based Markovian imitation, we didn’t exhaust every paper in the Markovian domain thus wasn’t aware of this one, but we are definitely awed by its performance against the expert. We will include the training curves of LS-IQ in the revision.
>
> > MCMC cost
>
> We add experiments to measure the time of posterior sampling, the time of prior MCMC sampling, and pure training time. As shown in Table 2, the posterior sampling is more consuming than prior sampling due to the additional computation of gradient of transition model. Importance sampling can bypass the additional cost since it involves only prior sampling.
>
> > Table 1
>
> The results are actually based on 5 seeds. We have updated the table to includ the standard deviation in the attached pdf file.
>
> > Goal position? Overimitation and extrinsic reward?
>
> It is more or less a philosophical question how humans understand the concept of “goal” through imitation. In L191, we stated that the “goal” means the last state $s_T$ in the sequence. The reviewer seems to have a different idea since in the cited psychology experiments, the object to be picked up at the end of demonstration sequences seems to be an extrinsic reward that further hints the “goal”.  We hope the reviewer would like to accept our hypothesis that agents may regard the last state as "goal" since it is simpler than the suggested one, and hence more preferable by the principle of Occam’s razor. Under this hypothesis, it is clear that overimitation is visiting the demonstrated goal with some causally unnecessary states in demonstrations.
>
> > non-Markovian rewards and Markovian states are usually sufficient?
>
> We hope to remind the reviewer of the distinction between sufficiency and necessity. Indeed, a non-Markovian decision-making problem is not necessarily represented by non-Markovian rewards and Markovian transitions. Our motivation for this design choice is communicated in L82-84.
>
> > Insight of Sec. 4
>
> As the reviewer points out, Sec. 4 proves there exists a reward function such that we can frame the MLE problem as a non-Markovian MaxEnt-RL problem, and this problem is solvable if MaxEnt-RL can be generalized to non-Markovian domains. This may further allude that the decision theory, especially offline MaxEnt-RL, might be a consequence of emergence from generative sequence modeling, rather than the essence.

---

### Official Review · Reviewer_LurP · 2023-07-07

**Soundness:** 3 good
**Presentation:** 2 fair
**Contribution:** 3 good
**Rating:** 6
**Confidence:** 2

**Summary:**

The paper proposes a generative model for learning non-Markovian decision-making from state-only sequences, where the policy is an energy-based prior in the latent space of the state transition generator. To solve the problems, the authors develop a maximum likelihood estimation method (LanMDP) for model-based imitation learning, which involves MCMC and importance sampling techniques. Besides, the papers shows how the learned model can be used for decision-making as inference, where model-free policy execution and model-based planning are equivalent to prior and posterior sampling, respectively. Experiments on a path planning task with non-Markovian constraints and several domains from the MuJoCo suite, LanMDP achieves comparable or superior performance than baselines (BC, LfO and GALfO).

**Strengths:**

State-only imitation is a important and a widely studied topic. This paper proposes an interesting idea of model-based imitation, and combines MCMC and importance sampling to estimate the model parameters from state-only data.
The paper demonstrates the versatility of the learned model for both model-free and model-based decision-making, where inference is performed by sampling from the prior or the posterior. The empirical evidence also supports the effectiveness of LanMDP.

**Weaknesses:**

I am not familiar with of the domain of nMDP, and I would like to share my concerns about the method and experiments:

My major concern is that the proposed method relies on MCMC and importance sampling techniques to estimate the model parameters and perform inference. However, these techniques can be computationally expensive, sample inefficient, and sensitive to initialization and tuning. Therefore, it is unclear how well the proposed method scales to more realistic and challenging scenarios.

Besides, the proposed method assumes that state-only sequences are sufficient to infer and imitate non-Markovian decision-making behaviors. In some cases, state-only sequences may not be enough to capture the relevant information or causal factors that influence decision-making, such as intentions, preferences, beliefs, or emotions. In such cases, state-only sequences is enough to learn meaningful or generalizable policies.


The motivation is unclear. For example, what is the advantage about using nMDP? Why does model-based imitation outperforms conventional methods? Additionally, I feel that there is a lack of cohesion between Sections. For example, Section 3 titles as 'Learning and Sampling'. However, it is separate from the previous Section and I do not know learning what and sampling what when I first read.

For experiments, the baseline methods are all from the 'imitation from observation' domain, which are built based on the standard MDP. I encourage authors to make comparisons with more related domain, i.e., methods that learn non-Markovian decision-making from state-only sequences.


**Questions:**

Here are some questions for the authors:

1. How do you evaluate the quality and diversity of the samples generated by the model? Do you use any quantitative metrics or qualitative visualizations to assess the fidelity and diversity of the generated sequences?

2. How do you compare the proposed method with other methods that learn non-Markovian decision-making from state-only sequences, such as inverse reinforcement learning or variational inference methods? What are the advantages and disadvantages of each approach?

**Limitations:**

The major limitations I am concerning are presented in 'Weakness' part.

---

> ### Author Rebuttal · Authors · 2023-08-10
>
> Thank you for your feedback!
>
> > Computational cost of MCMC.
>
> We recognize this concern. But we are rather optimistic in the short-run MCMC methods [1] since in realistic tasks the dimensionality of the action space is normally small in comparison with the state space. We add experiments to measure the time of posterior sampling, the time of prior MCMC sampling, and pure training time. As shown in Table 2, the posterior sampling is more consuming than prior sampling due to the additional computation of gradient of transition model. Importance sampling can bypass the additional cost since it involves only prior sampling.
>
> > ​​Assumes that state-only sequences are sufficient to infer and imitate non-Markovian decision-making behaviors?
>
> We feel that this is a misunderstanding. We hope the reviewer would like to clarify what “sufficient” means. If it means state-only sequences are insufficient to uniquely identify a latent action, we certainly agree. Actually, existing works attribute the unsatisfying performance of inverse dynamics methods to such insufficiency [2]. We indeed take this into consideration in model design by making the policy multi-modal.
>
> In terms of “intentions,... emotions”, we want to argue that people don’t, for example, carry a belief label when behaving, but we can infer it from our observations. Therefore, these are all latent variables, just like the latent actions in our model.
>
> > The motivation is unclear.
>
> The motivation is communicated in the introduction. Specifically, we want to draw the reviewer’s attention to L28-29, where we reveal the limitation of the conventional imitation method due to the reliance on Markovian assumption and TD learning. The motivation is further illustrated with a toy task in Sec. 5.1. We hope the reviewer would like to discuss with reviewers W5w2 and wvit the relevance of nMDP.
>
> We believe “model-based imitation outperforms conventional methods” is a misconception of our results. For example, BCO in Fig. 3 is also a model-based method. Our employment of a model-based method is largely due to the restrictions of latent actions and nMDPs, where model-free methods may not directly apply. We reckon that this misconception may attribute to the statement “We develop maximum likelihood estimation to achieve model-based imitation” in the abstract (L8), for it implies certain senses of purpose towards model-based imitation despite our real intention of conveying all constituents of nMDP (L6) are learned. We will resolve this confusion in the revision.
>
> > A lack of cohesion between Sections.
>
> Thanks for this feedback! We believed the structure of this paper was implied in the last two paragraphs (L36-62) of the introduction, and the format of “modeling, learning, analysis” was conventional in machine learning papers. We will add a paragraph at the end of the introduction to explicitly sketch the structure in the revision. In short, in Sec. 2 we introduce the model; in Sec. 3 we introduce learning this model with neural networks as function approximators, which involves sampling; in Sec. 4 we analyze the connections between the learned model with sequential decision-making problems.
>
> > Comparisons with methods that learn non-Markovian decision-making from state-only sequences.
>
> To the best of our knowledge, there is no established research in this particular setting.
>
> > Quantitative metrics or qualitative visualizations for the fidelity and diversity of the generated sequences?
>
> The quality of generated sequences in the cubic planning task is evaluated by the residual errors and acceptance rate detailed in L233-239. In a word, we evaluate if the model learns to generate cubic sequences. This can also be qualitatively evaluated by the figures we plot in Fig. 2. For Mujoco tasks, the quality is assessed by the final score of the test trajectory. To answer the reviewer’s request, we upload some videos for qualitative evaluation.
>
> Thanks for raising our attention to diversity, for which we don’t have a systematic metric currently. In a published benchmark for unsupervised RL [3], such a measure is also omitted. Other sequence models for decision-making don’t have such evaluation either. We agree this is an important concern and will include this limitation in the revision.
>
> > Comparison with as IRL or variational inference methods?
>
> We hope the reviewer has been reminded in previous answers that there seems to be no established work for non-Markovian IRL yet due to IRL’s reliance on the Markovian limitation. We pick behavior cloning (BC) as the major baseline because it is free from the Markovian assumption. Putting the Markovianness aside, we believed with abundant data, behavior cloning with action labels empirically upper bounds the performance of IRL. This is supported by the result from [4,5], as well as our results in FIg. 3. We are very much anticipating future progress from RL’s perspective.
>
> We feel thrilled for the reviewer mentioned variational inference as an alternative to MCMC. It is actually the next project in our plan, as we believed learning amortized sampler for latent EBM in nMDP deserved a standalone paper.
>
>
> [1] Erik Nijkamp, Mitch Hill, Song-Chun Zhu, and Ying Nian Wu. Learning non-convergent non-persistent short-run mcmc toward energy-based model. Advances in Neural Information Processing Systems, 32, 2019.
>
> [2] Zhu, Zhuangdi, et al. "Off-policy imitation learning from observations." Advances in Neural Information Processing Systems 33 (2020): 12402-12413.
>
> [3] Laskin, Michael, et al. "URLB: Unsupervised reinforcement learning benchmark." Neural Information Processing Systems Track on Datasets and Benchmarks (2021).
>
> [4] Goo, Wonjoon, and Scott Niekum. "Know Your Boundaries: The Advantage of Explicit Behavior Cloning in Offline RL." (2022).
>
> [5] Florence, Pete, et al. "Implicit behavioral cloning." Conference on Robot Learning. PMLR, 2022.

---

> > ### Comment · Reviewer_LurP · 2023-08-16
> >
> > Thanks for your detailed response. I have carefully read the response and authors' discussion with other reviewers (especially with Reviewer W5w2). I think the response addresses my major concerns about computational cost and the motivation of the paper. I feel more positively about the paper and would like to raise my score to 6.

---

> > > ### Author Response · Authors · 2023-08-18
> > >
> > > Thank you very much!

---

### Official Review · Reviewer_N9u8 · 2023-07-13

**Soundness:** 3 good
**Presentation:** 3 good
**Contribution:** 3 good
**Rating:** 7
**Confidence:** 2

**Summary:**

The goal of this approach is to learn from state-only sequences, in the case where action labels are not present, especially in non-markovian settings. The method formulates a non-Markovian Decision Process (nMDP) with latent actions as a Latent-space Energy Based Model, showing that the inference on the model is equivalent to policy execution/planning. Using a dynamics model and posterior sampling, the learned model can be used for planning, even for reaching new goal states. The paper alsow draws a connection of the proposed approach to max-ent RL. The results show good performance for non-markovian (and even some markovian) tasks.

**Strengths:**

- This paper addresses an interesting problem of learning in non-Markovian settings
- The application to learning from action-free data is very interesting as well
- The approach outperforms IRL and BCO baselines on toy and control tasks
- The approach is sound and novel (to my knoweldge)


**Weaknesses:**

I think there could be more analysis on the different types of non-Markovian tasks (i.e. tasks that require more long term versus short term memory). In general, more robotics focused applications would be a better showcase for this method. The context seems to be important - it would be good to investigate howhe use of sequence based architectures such as the Decision Transformer (Chen et al., 2021) fits into this general problem as well.

**Questions:**

See weaknesses section

**Limitations:**

These are sufficiently addressed

---

> ### Author Rebuttal · Authors · 2023-08-10
>
> Thank you very much for the feedback!
>
> > I think there could be more analysis on the different types of non-Markovian tasks (i.e. tasks that require more long term versus short term memory). In general, more robotics focused applications would be a better showcase for this method. The context seems to be important - it would be good to investigate howhe use of sequence based architectures such as the Decision Transformer (Chen et al., 2021) fits into this general problem as well.
>
> We agree that systematically categorizing long-/short-term memory in nMDPs and testing various sequential decision-making architectures [1,2] in different regimes are crucial research topics. However, to the best of our knowledge, research as such awaits more infrastructure due to the lack of systematic benchmarks. We hope our work can invoke more efforts in the emerging area of non-Markovian decision-making, which could get us more prepared for the suggested investigation of the influence of different sequence-based architectures. We’d also love to explore in futhre work more robotics-focused applications beyond the toy or simulated control tasks presented here.
>
> [1] Janner, Michael, Qiyang Li, and Sergey Levine. "Offline reinforcement learning as one big sequence modeling problem." Advances in neural information processing systems 34 (2021): 1273-1286.
>
> [2] Ajay, Anurag, et al. "Is conditional generative modeling all you need for decision-making?." arXiv preprint arXiv:2211.15657 (2022).

---

### Official Review · Reviewer_tbAW · 2023-07-28

**Soundness:** 3 good
**Presentation:** 1 poor
**Contribution:** 2 fair
**Rating:** 3
**Confidence:** 4

**Summary:**

This paper proposes a new approach to imitation learning from observation (ILfO) on a non-Markovian decision process (nMDP). To achieve this, the authors derive their own objective using maximum likelihood estimation, drawing inspiration from deep generative modeling of state-only sequences. After providing a connection between probabilistic inference and decision-making, the authors conduct experiments on cubic curve planning and MuJoCo control tasks.

**Strengths:**

While there are several works addressing ILfO for MDPs, this work appears to be the first attempt to handle ILfO for nMDPs. To this end, the authors derive their own objective based on probabilistic inference.

**Weaknesses:**

**[Hard to understand]** this paper is hard to understand due to the lack of explanation. For example, there are various policy-like terms as $p\_\theta(A|S)$, $p\_alpha(a\_t|s\_{0:t})$, and $p\_\theta(a\_t|s\_{0:t+1})$. I believe $p\_\alpha(a\_t|s\_{0:t})$ corresponds to the policy of the current agent in RL. However, in the context of RL, there is no explicit explanation provided for what $p\_\theta(A|S)$ and $p\_\theta(a\_t|s\_{0:t+1})$ correspond to. In addition, the final objective function is not formulated.

**[Lack of baselines]** In the context of RL, there are two ways to improve sample efficiency: off-policy RL and model-based RL. Thus, at least one of the off-policy ILfO algorithms (e.g., OPOLO [1]) and one of the model-based ILfO algorithms (e.g., MobILE [2]) should be considered as baselines.

**[Objective]** First of all, I wonder if the proposed objective (5) is valid in a stochastic environment. Indeed, the learned $p\_\beta$ can differ from the true transition in a stochastic environment, although it is possible to learn an accurate transition with enough online samples. Moreover, (8) may also allow for learning an accurate transition with enough online samples, but it is modified without any theoretical justification from the original objective (5).

**[Exploration]** I think LanMDP might fail to obtain a good policy in a stochastic environment (i.e., with stochastic transition probabilities) because there is no exploration. For example, if the algorithm obtains $(s,a,s’)$ during interaction, then there is no need to find other possible actions $a’$ that might be better to reach $s’$ from $s$.

**[Motivation of task]** This work aims to solve ILfO for nMDPs. However, there are no motivating examples explaining why ILfO for nMDPs should be studied. Specifically, even though the reward function is non-Markovian, the expert policy should not necessarily be a non-Markovian expert policy. Moreover, while the expert policy is non-Markovian, it is not clear if existing ILfO algorithms fail to solve ILfO for nMDPs.

**[Motivation of LanMDP]** It is not clear why we should use decision making as inference instead of a simple modification of prior works. Indeed, we can also handle ILfO on an nMDP by modifying MobILE - for example, if we replace $(s,a)\sim\mathbb{E}\_{d^\pi}$ and $s\sim D\_e$ with $\tau\sim\mathbb{E}\_{\pi}$ and $\tau\_s\sim D\_e$ for a non-Markovian policy $\pi$ and a state-only trajectory $\tau\_s$, then it can be used to solve ILfO on an nMDP.

[1] Zhu, Zhuangdi, et al. "Off-policy imitation learning from observations." Advances in Neural Information Processing Systems 33 (2020): 12402-12413.

[2] Kidambi, Rahul, Jonathan Chang, and Wen Sun. "Mobile: Model-based imitation learning from observation alone." Advances in Neural Information Processing Systems 34 (2021): 28598-28611.

**Questions:**

Mentioned in the weakness + following questions

- (related to [Lack of baselines]) In experiments, I believe that at least one of the off-policy ILfO algorithms (e.g., OPOLO) and one of the model-based ILfO algorithms (e.g., MobILE) should be added as baselines.
- (related to [Exploration]) Compared to MobILE, why does LanMDP not need to consider an exploration term? In addition, is there no risk in a stochastic environment without exploration such as mentioned in weakness part?
- What is the final objective of LanMDP, which is used for gradient-based updates? Additionally, compared to the baselines (OPOLO and MobILE), what is the main contribution of LanMDP?
- I think $R: S^+\rightarrow \mathbb{R}$ is a generalization of $R(s)$. Is there any reason not to use $R: S^+\times A^+\rightarrow \mathbb{R}$, which is a natural generalization of $R(s,a)$?
- In equation (13), we need to compute $c(a\_t;s\_{0:t+1})$, which involves an expectation over $p\_\alpha I believe this part involves intractable integration in the continuous domain. If the authors use MCMC to address this issue, I am curious about the computational time it takes (compared to other baselines).

**Limitations:**

.

---

> ### Author Rebuttal · Authors · 2023-08-10
>
> Thank you very much for your feedback!
>
> > [Hard to understand] this paper is hard to understand due to the lack of explanation. For example, there are various policy-like terms as $p_\theta(A|S)$, $p_\alpha(a_t|s_{0:t})$, and $p_\theta(a_t|s_{0:t+1})$. I believe $p_\alpha(a_t|s_{0:t})$ corresponds to the policy of the current agent in RL. However, in the context of RL, there is no explicit explanation provided for what $p_\theta(A|S)$ and $p_\theta(a_t|s_{0:t+1})$ correspond to.
>
> We believe we have provided sufficient explanation for every notation we use. For example, in L110-111, we explain that $p_\theta(A|S)$ denotes $p_\theta(a_{0:T-1}|s_{0:T})$, a posterior probability in which $A$ denotes the complete action sequence $a_{0:T-1}$, $S$ denotes the complete state sequence $s_{0:T}$. We understand that this posterior probability isn’t usually seen in the literature of RL. So we explicitly define $p_\theta(a_{0:T-1}|s_{0:T})$ in Eq. 4. The notation of $p_\theta(a_t|s_{0:t+1})$ also follows the conventions of conditional distribution and Bayesian statistics.
>
> > [Lack of baselines] In the context of RL, there are two ways to improve sample efficiency: off-policy RL and model-based RL. Thus, at least one of the off-policy ILfO algorithms (e.g., OPOLO [1]) and one of the model-based ILfO algorithms (e.g., MobILE [2]) should be considered as baselines.
>
> We appreciate the reviewer's suggestion to include off-policy and model-based ILfO algorithms as baselines. We initially considered MobILE as a baseline since it is also a model-based method for imitation from observation. However, our experiments with MobILE, using various sets of expert demonstration trajectories, did not yield satisfactory results. We will include this observation in the revision. We have included the result of OPOLO in Table 1 in the attached PDF file. The proposed LanMDP shows comparable performance.
>
> > [Objective] First of all, I wonder if the proposed objective (5) is valid in a stochastic environment. Indeed, the learned $p_\beta$ can differ from the true transition in a stochastic environment, although it is possible to learn an accurate transition with enough online samples. Moreover, (8) may also allow for learning an accurate transition with enough online samples, but it is modified without any theoretical justification from the original objective (5).
>
> The learning paradigm we employ is maximum likelihood. Eq. 5 is the general maximum likelihood objective, which is supposed to be applicable to any distribution to be learned, no matter whether the environment is deterministic or stochastic. The potential issue of using Eq. 5 alone is that the learned transition may not be grounded because actions are latent. To overcome this issue, we introduce the extra online learning term in Eq. 8, making it deviate from Eq. 5. Apparently, it is derived from jointly optimizing two log-likelihood, Eq. 5 and $L_{online}(\beta) = \sum_{i=1}^{m} \sum_{t=1}^{T} \log p_\beta (s_{t+1}^i | s_{t}, a_t)$. We will include this explanation in the revision.
>
> > [Exploration] I think LanMDP might fail to obtain a good policy in a stochastic environment (i.e., with stochastic transition probabilities) because there is no exploration. For example, if the algorithm obtains $(s, a, s’)$ during interaction, then there is no need to find other possible actions a’ that might be better to reach s’ from s.
>
> We disagree with this comment. As stated in L182-184, LanMDP is inherently advantageous in exploration thanks to the energy-based modeling of policies. The maximum entropy inclination in energy-based policies has been shown to encourage explorations in [1].
>
> > [Motivation of task] This work aims to solve ILfO for nMDPs. However, there are no motivating examples explaining why ILfO for nMDPs should be studied. Moreover, while the expert policy is non-Markovian, it is unclear if existing ILfO algorithms fail to solve ILfO for nMDPs.
>
> We could not help arguing that this critic is unsubstantiated. Actually, the cubic planning task in Sec 5.1 is deliberately included to motivate the importance of non-Markovian modeling. The empirical results justify the insufficiency of the Markovian Imitation Learning methods (with or without action labels) and the efficacy of the proposed model.
>
> > [Motivation of LanMDP] It is not clear why we should use decision making as inference instead of a simple modification of prior works. Indeed, we can also handle ILfO on an nMDP by modifying MobILE.
>
> As far as we know, there is no established work showing the suggested modification can principally solve the problem of imitation learning in non-Markovian domain. We also believe the non-Markovian extension of state-matching methods is non-trivial. It would be helpful if the reviewer would like to provide us with some pointers.
>
> > I think $R: S+ -> \mathbb{R}$ is a generalization of $R(s)$. Is there any reason not to use $R: S+ x A+ -> \mathbb{R}$, which is a natural generalization of $R(s, a)$?
>
> In Sec 2, we introduce statistical assumptions in LanMDP. Since actions are assumed to be unobservable in the dataset, we exclude them from the dependency of $R$. It could be an interesting future direction to explore the setting that the reviewer suggested.
>
> > In equation (13), we need to compute $c(a_t; s_{0:t+1})$, which involves an expectation over $p_\alpha$ I believe this part involves intractable integration in the continuous domain. If the authors use MCMC to address this issue, I am curious about the computational time it takes (compared to other baselines).
>
> We add experiments to measure the time of posterior sampling, the time of prior MCMC sampling, and pure training time in Table 1 in the attached PDF. Importance sampling can bypass the additional cost of posterior sampling since it involves only prior sampling.
>
> [1] Haarnoja, Tuomas, et al. "Reinforcement learning with deep energy-based policies." International conference on machine learning. PMLR, 2017.

---

> > ### Comment · Reviewer_tbAW · 2023-08-16
> >
> > Thank you for the response. However, I still have the following questions.
> >
> > 1. **Notation**
> >
> >     I want to clarify that I understand the definition of $p\_\theta(A|S)$. However, what I am trying to convey is this: In Equation (2), the authors establish the definition of $p\_\theta$ using $p\_\alpha$ and $p\_\beta$. This may correspond to the prior $\times$ likelihood in Bayesian perspective. Thus, the definition of $p\_\alpha(a\_t|s\_{0:t})$ is given in Equation (3). Given this context, my question is: What is the definition of $p\_\theta(a\_t|s\_{0:t+1})$, which may correspond to the posterior, as presented in Equation (10)? Is it really independent of $s\_{t+2:T}$ and reward terms?
> >
> >     I have this question because the authors employ the assumption of Markovian transition to establish the final equality in Equation (10) (i.e., $p\_\theta(a\_t|s\_{0:T})=p\_\theta(a\_t|s\_{0:t+1})$). However, my understanding is that in order to derive this final equality, the **trajectory should be independent of the reward**, given that rewards do not adhere to the Markovian property. However, assuming that trajectories are sampled without observing reward is strange. The authors assume that expert demonstrations maximize this non-Markovian reward, while the learned policy aims to mimic these expert demonstrations. Consequently, the trajectory sampled from the learned policy is inherently dependent on the reward. I am not sure if this claim is accurate, but my intention is to emphasize the **need for a more detailed derivation in order to obtain Equation (10)**. If the authors want to claim that Equation (10) still holds for trajectories dependent on the reward, then it should be derived theoretically.
> >
> > 2. **Baseline**
> >
> >     ~~I cannot find the performance of OPOLO in Table 1. In my understanding, Table 1 in the paper compares BC and LanMDP with different context lengths, and Table 1 in the appendix provides the hyperparameters. Could you please provide more specific information about the location of the experimental results of OPOLO in Table 1?~~ Apologies for the confusion. I found Table 1 in the attached file. Based on these results, I believe even more strongly that an additional realistic experiment, perhaps a case that reflects ILfO in nMDP, is required to justify LanMDP.
> >
> > 3. **Objective**
> >
> >     The provided explanation is not enough. What I want to convey is the theoretical justification for including $L\_\text{online}$ within $L(\theta)$ for both model and policy learning. Can we be certain that the joint optimization theoretically converges to the desired model and policy? Does this fundamentally resolve the issue of ungrounded transitions with respect to arbitrary $w\_\beta$? If not, there must be more theoretical derivations.
> >
> > 4. *Exploration* (Minor)
> >
> >     I missed details in L182-184 and I appreciate your pointing that out. I am still confused about when LanMDP interacts with the environment and when the replay buffer is used. Based on my understanding, LanMDP interacts with the environment initially and during training, particularly in the 'Use energy-based policy with $\alpha\_0$ ...' part and the '**Transition learning**: Update replay buffer with trajectories from the current policy model ...' part in the Algorithm in Appendix A.  Additionally, the replay buffer is utilized to compute $L\_\text{online}(\beta)$. If my understanding is accurate, then why is LanMDP more sample efficient than off-policy ILfO, even though it does not sample trajectories from the learned model? Additionally, I would like to understand how the authors utilize the ensemble, as mentioned in L183 of the Appendix.
> >
> > 5. **Motivation**
> >
> >     I disagree that the cubic planning task serves as a motivating example. The cubic planning task actually motivates the proposed algorithm to tackle the nMDP, rather than serving as the motivation for the ILfO in the nMDP. To claim that ILfO in the nMDP is truly valuable, it must correspond to real-world problems. If this paper considers ILfO in POMDP, then I think it is valuable, but I cannot find any realistic examples that reduce to the ILfO in the nMDP. Additionally, MobILE can be applied simply to solve ILfO in the nMDP by using $\pi(a\_t|s\_{0:t})$ instead of $\pi(a\_t|s\_t)$ (and in Equation (1) of the MobILE paper, consider using $f(s_{0:t})$ with $(s_{0:t},a_{0:t})\sim D_{\hat{P}}^pi$ and $(s_{0:t})\sim D_e$ instead of $f(s)$, if needed). Compared to this ad-hoc approach, what is the main advantage of LanMDP, which also constructs the objective (8) without theoretical justification?

---

> > > ### Comment · Reviewer_tbAW · 2023-08-16
> > >
> > > Apologies for the confusion. I found Table 1 in the attached file. Based on these results, I believe even more strongly that an additional realistic experiment, perhaps a case that reflects ILfO in nMDP, is required to justify LanMDP.

---

> > > ### Author Response · Authors · 2023-08-18
> > >
> > > Thanks for helping us understand your confusion.
> > >
> > > 1. **Notation and reward independence?**
> > >
> > > It seems there is still a misunderstanding. $p_\theta(a_t|s_{0:t+1})$ is the posterior $\frac{p_\alpha(a_t|s_{0:t})p_\beta(s_{t+1}|s_t, a_t)}{p_\theta(s_{t+1}|s_{0:t})}$, in which we use subscript $\theta$ whenever both $\alpha$ and $\beta$ are involved, as defined in L95. We omitted the derivation of Eq. (10) because we believed it was apparent under conditional independence described in our probabilistic graphical model:
> > >
> > > $p_\theta(a_t|s_{0:T}) = \frac{p_\theta(a_t, s_{0:T})}{p_\theta(s_{0:T})} = \frac{p_\theta(a_t, s_{0:t})p_\beta(s_{t+1}|s_t,a_t)p_\theta(s_{t+2:T}|s_{0:t+1})}{p_\theta(s_{0:t})\int p_\alpha(a_t|s_{0:t})p_\beta(s_{t+1}|s_t,a_t)da_t p_\theta(s_{t+2:T}|s_{0:t+1})} = \frac{p_\alpha(a_t|s_{0:t})p_\beta(s_{t+1}|s_t, a_t)}{p_\theta(s_{t+1}|s_{0:t})}=p_\theta(a_t|s_{0:t+1}) $.
> > >
> > > In the derivation above, we don’t have explicit variables for reward or return because the expert policy is assumed to have optimized them – they are marginalized variables in the policy distribution. Marginalization, however, does not mean independence.
> > >
> > > 2. **Baseline**
> > >
> > > We believe the additional experiments do not rule out our assumption that BC empirically upper bounds the performance of existing methods in Markovian domains, especially in tasks with high-dimensional actions. In non-Markovian domains, BC is still a sufficient baseline since there aren't established ILfO methods. But we agree that the suggested investigation can be an interesting future work.
> > >
> > > 3. **Objective and theoretical justification**
> > >
> > > We understand your concerns in jointly optimizing two log-likelihoods (in our case, one marginal likelihood and one conditional likelihood). We are also aware of the emerging literature on RL theory, especially model-based RL, which analyzes the convergence of jointly learning a transition model and a policy. However, we hope to remind the reviewer that intellectual efforts are needed in RL theory partially because its proof of convergence cannot be trivially reduced to maximum likelihood estimation, whose asymptotic theory has been established long ago. Formally speaking, MLE is *asymptotically consistent*, in the sense that if the data were generated by the modeled function family (which fulfills regularity conditions of identifiability, compactness, continuity, and dominance) and we have a sufficiently large number of observations, then it is possible to find the desired model parameters $\theta$ with arbitrary precision [1]. Multiplying a collection of likelihoods is called composite likelihood [2]. As long as all components satisfy the regularity conditions, the asymptotic consistency still holds.
> > >
> > > [1] Maximum likelihood estimation. (2023, August 9). In Wikipedia.
> > >
> > > [2] Varin, Cristiano, Nancy Reid, and David Firth. "An overview of composite likelihood methods." Statistica Sinica (2011): 5-42.
> > >
> > > 4. **Online interaction**
> > >
> > > We never claim that LanMDP is more sample-efficient than off-policy ILfO.
> > >
> > > We use an ensemble to stabilize gradients from importance sampling. Specifically, we hold two transition models $f_1, f_2$ trained from different initialization and data shuffling. During importance sampling, a small portion of action samples with significant disagreement in the predictions of $f_1, f_2$ are discarded. Empirically, we find this trick very effective in stabilizing neural network training. Note that this ensemble is not used in online interaction.
> > >
> > > 5. **Motivating experiments and twisted MobILE**
> > >
> > > We believe cubic planning is an abstraction of some realistic tasks. For example, in autonomous driving, cubic Bezier curve planning, which involves 3-step lookahead, plays a crucial role in making the path smoother. Our experiments show that only when the non-Markovian context is sufficiently represented, we can learn a policy that does not require any lookahead during execution.
> > >
> > > While we agree that MobILE is an awesome work, we hope our work can be evaluated as self-contained research. At the level of formulation, the framework of distribution matching in MobILE is an alternative to maximum likelihood estimation in LanMDP, in which any concrete modeling of the distribution-to-be-matched deserves a standalone study. An ad-hoc twist of this rigorous method towards non-Markovianness, were it to lead to wide empirical success, reveals unnecessary assumptions in the twisted method, rather than invalidating other formulations. At the level of specifics, we would like to remind the reviewer that (1) MobILE implicitly assumes the policy is unimodal, see their footnote 2 and 3; and (2) their reward-based formulation leads to the requirement of model-based policy search, which always involves either Monte Carlo sampling or TD learning. In contrast, LanMDP (1) adopts multimodal energy-based policies, making the two footnotes less concerning; (2) bypasses model-based policy search with posterior sampling/importance sampling.

---

> > > > ### Comment · Reviewer_tbAW · 2023-08-19
> > > >
> > > > Thank you for your response.
> > > > While I do have additional questions, my primary reason for disagreeing with accepting this paper is related to its motivation.
> > > > Therefore, I will focus on discussing the motivation due to the lack of discussion time.
> > > >
> > > > I have two questions about the problem setting - ILfO in nMDP.
> > > >
> > > > 1. Is this setting (i.e., ILfO in nMDP) really useful in practice?
> > > >
> > > >     Although the authors claim that the cubic curve planning task is realistic, I believe that in many real-world cases, POMDPs can be converted into MDPs with latent state spaces, rather than nMDPs.
> > > >
> > > > 2. Is it possible for LanMDP to obtain a better solution than ILfO in MDPs, such as OPOLO?
> > > >
> > > >     In the cubic curve planning task, does LanMDP still perform well with a context length greater than 4? In the paper, the authors mentioned that 6 is excessive expressivity. However, there is no apparent reason why LanMDP with a context length greater than 4 would perform worse than LanMDP with a context length of 4. If LanMDP fails to obtain a good policy with a context length greater than 4, is it due to 1) ILfO in nMDP being an ill-posed problem requiring additional information to solve, or 2) is it simply because LanMDP is sensitive to context length, implying the impracticality of the algorithm?
> > > >
> > > > In addition, the reason I inquired about the objective and theoretical justification is that the authors do not seem to address certain issues in ILfO, like inverse dynamics disagreement [1]. Previous ILfO algorithms have considered inverse dynamics disagreement, which arises due to the absence of expert action. While the authors introduce $L_\text{online}$ to learn the model, there is no theoretical reason to add this term and it appears that this algorithm does not explicitly address this issue. Considering the potential instability of joint optimization, I am concerned that ILfO-specific problems might exacerbate this instability. To handle ILfO in nMDP, I believe theoretical analysis is necessary. For example, minimizing (8) is equivalent to optimizing a desirable ILfO objective, and it represents a fundamental approach compared to simple extensions of OPOLO or previous ILfO algorithms.
> > > >
> > > > For example of naive extension of OPOLO, why not simply define a new MDP? For example, for a context length of 2, we can define $s^2_t=(s_t, s_{t-1})$ and $s^2_{t+1}=(s_{t+1},s_t)$, where $s_{t+1}\sim T(\cdot|s_t,a_t)$, and then run OPOLO for this MDP. I would be inclined to agree that LanMDP can outperform naive extensions of ILfO if there are theoretical derivations or, at the very least, practical experiments to support this claim. In addition, considering that this extension of OPOLO and LanMDP uses concatenated states as inputs for the policy, I believe both approaches might face challenges in high-dimensional tasks.
> > > >
> > > > [1] Yang, Chao, et al. "Imitation learning from observations by minimizing inverse dynamics disagreement." Advances in neural information processing systems 32 (2019).

---

> > > > > ### Comment · Reviewer_tbAW · 2023-08-19
> > > > >
> > > > > Furthermore, the expert demonstrations in MuJoCo exhibit poor performance. Why not consider using better expert demonstrations, such as those from D4RL or training an expert policy using SAC, which has shown good performance in MuJoCo? Since the authors use performance as the metric for imitation learning, it should truly reflect the optimal performance.

---

> > > > > ### Author Response · Authors · 2023-08-21
> > > > >
> > > > > 1. **LanMDP vs POMDP**
> > > > >
> > > > > As discussed in L76-88, while POMDP highlights the Markovianness in latent states that might be absent in observations, **LanMDP shifts attention towards the Markovianness in the transition and potential non-Markovianness in the reward/policy**. As transition and policy correspond to materiality/physics and intentionality/psychology respectively, their temporal discordance is conjectured to be a pivotal driver for human minds to disentangle the corresponding conceptual domains. This hypothesis, we believe, harmonizes effectively with the intuitive experiences of contemporary humans.
> > > > >
> > > > > For instance, classical mechanics, as described by Newton's laws of motion and Hamiltonian mechanics, often involves systems with behavior where the next state is entirely determined by the current state and the forces acting upon it. Nevertheless, an individual's human decision-making process does not invariably adhere to these mechanistic Markovian dynamics, as shown in the cited psychology experiments [1]. We believe such discordance is better exposed with our model, which bears the potential to bestow a fresh perspective upon ongoing research into generative sequence models and the emergence of intelligence.
> > > > >
> > > > > 2. **LanMDP with excessive expressivity**
> > > > >
> > > > > The critique suggesting that LanMDP might suffer from the issue of excessive expressivity **lacks substantiation.** Contrary to this assertion, our discourse unequivocally addresses LanMDP’s capability of handling excessive expressivity in Fig. 2(e)(f) and L258, as well as L295-296. It is also clear in the additional experiment that LanMDP shows performance comparable to OPOLO in MDPs.
> > > > >
> > > > > 3. **Theoretical justification**
> > > > >
> > > > > Although we don’t intend to claim credit for the theory of MLE, which many pioneering statisticians have established, we firmly believe that our approach should not be unjustly deemed as lacking theoretical justification. We hope our point-to-point answers below, together with our prior responses, can help deepen the reviewer’s understanding of our method.
> > > > >
> > > > > - $L_{online}$ in the objective: This second likelihood helps ground the transition with online data. The convergence analysis of the composite objective is covered in the asymptotic theory of composite likelihood, as cited in the previous response.
> > > > > - Inverse dynamics disagreement: Actually, $D_{KL}(p_\theta(a | s, s') | p^*(a| s, s'))$ is minimized in our maximum composite likelihood. The suggested paper [2] (1) assumes deterministic dynamics and (2) minimizes an upper bound of this KL divergence, whose general validity is in doubt. The MLE in our method, which is entirely different from [2],  leverages online data to consistently estimate $p_\beta(s'|s, a)$. Then $p_\alpha(a|s)$ will be estimated consistently. As a result, LanMDP automatically estimates $p_\theta(a|s, s') \propto p_\alpha(a|s)p_\beta(s'|s, a)$ consistently.
> > > > > - Joint optimization: Joint optimization is a widely adopted practice in modern deep learning. See the success of model-based RL [3].
> > > > >
> > > > > 4. **OPOLO with concatenated states**
> > > > >
> > > > > Defining a new state space by concatenating the ground-truth length of states is not a practical solution even though it may help reduce the non-Markovianness. This is because **usually, we don’t have a priori knowledge of the sufficient context length**. Our experiments with excessive expressivity, as highlighted above, illustrate our model’s capability in discovering sufficiency albeit having excessive power. OPOLO, as a TD-learning method with a Markovian Bellman operator, is inherently incapable of such learning [4].
> > > > >
> > > > > **At last, we would like to express gratitude for this detailed discussion. We earnestly trust that the reviewer will acknowledge the concerns we have addressed.**
> > > > >
> > > > > [1] Horner, Victoria, and Andrew Whiten. "Causal knowledge and imitation/emulation switching in chimpanzees (Pan troglodytes) and children (Homo sapiens)." Animal cognition 8 (2005): 164-181.
> > > > >
> > > > > [2] Yang, Chao, et al. "Imitation learning from observations by minimizing inverse dynamics disagreement." Advances in neural information processing systems 32 (2019).
> > > > >
> > > > > [3] Hafner, Danijar, et al. "Dream to control: Learning behaviors by latent imagination." ICLR 2020 (2019).
> > > > >
> > > > > [4] Ferrer-Mestres, Jonathan, et al. "Solving k-mdps." Proceedings of the International Conference on Automated Planning and Scheduling. Vol. 30. 2020.

---

> > > > > > ### Comment · Reviewer_tbAW · 2023-08-22
> > > > > >
> > > > > > 1. Usually, we use MDP when we believe we have all the state information, and if not, we use POMDP. However, in my opinion, it's not common to use nMDP. For example, in the context of driving a car, a more natural setting would be state=(position, velocity) and action=acceleration. That's why I consider cubic planning as a good synthetic task rather than a realistic one. What I'm really interested in is which **realistic and widely applicable** tasks are naturally considered as nMDPs?
> > > > > >
> > > > > > 2. I think with enough large context length, LanMDP should perform well. So I'm just curious about the results for **cubic planning with LanMDP using a context length of 6 (or more than 6)**, similar to what's shown in Fig 2(c). This may have already been conducted by the authors, as indicated by the results in Fig 2(e)(f).
> > > > > >
> > > > > > 3. What I want to emphasize is the consideration of IDM, rather than making a direct comparison with the reference. While joint optimization will converge, the question is, where does it converge? Since two terms in Equation (8) optimize the same neural network, we **cannot be certain of the convergence point without theoretical derivations**. In addition, I wonder if IDM may amplify such phenomena. Thus, without rigorous theoretical derivation, it's unclear why LanMDP is superior to a heuristic modification of previous ILfO methods.
> > > > > >
> > > > > > 4. I believe that pointing out the lack of prior knowledge of sufficient context length is *not an appropriate response*. LanMDP also treats context length as a hyperparameter. Therefore, modified OPOLO can use the same policy structure as LanMDP, **based on the new MDP defined as $s^\text{new}_t=(s\_{0:t})$**. Could you provide an example problem where modified OPOLO is insufficient, but LanMDP should be used?

---

> > > > > > > ### Author Response · Authors · 2023-08-22
> > > > > > >
> > > > > > > Although we acknowledge the significance of adhering to the official deadline, we find it imperative to address certain misconceptions put forth by the reviewer at this juncture.
> > > > > > >
> > > > > > > 1. nMDP: All state spaces, as long as they are distinct from the observation spaces, are more or less defined by researchers. The latent states that generate the observations can either be an MDP as in POMDP, or an nMDP as in LanMDP -- **it is just a design choice.** We hope the reviewer is aware that there might be historical reasons that cause other models to be less popular than POMDP. As discussed in L32-33 in the main text, we believe the emerging trend of exploring other models can be deduced from the fact that an analysis of the limitation of Markovian rewards [1] won the best paper award in NeurIPS 2021.
> > > > > > >
> > > > > > > 2. Context length of 6: The requested result has been included in **Fig. 1 in the supplementary**.
> > > > > > >
> > > > > > > 3. Convergence point: As stated in our previous response, **the "consistency" of MLE means convergence to the ground-truth distributions, as long as they lie in the assumed function families** [2].
> > > > > > >
> > > > > > > 4. Excessive expressivity in TD-learning: Though LanMDP treats the context length as a hyperparameter, it does not need to know the ground-truth length. **LanMDP's success in experiments with excessive expressivity implies that we can use LSTM or other sequence architectures to get rid of manually controlling context lengths**. Vanilla TD-learning methods with concatenated states, however, don't appear to be capable of discovering proper historic contexts, as shown in the paper we cited in our previous response [3].
> > > > > > >
> > > > > > > [1] Abel, David, et al. "On the expressivity of markov reward." Advances in Neural Information Processing Systems 34 (2021): 7799-7812.
> > > > > > >
> > > > > > > [2] Maximum likelihood estimation. (2023, August 9). In Wikipedia.
> > > > > > >
> > > > > > > [3] Ferrer-Mestres, Jonathan, et al. "Solving k-mdps." Proceedings of the International Conference on Automated Planning and Scheduling. Vol. 30. 2020.

---

### Author Rebuttal · Authors · 2023-08-10

We would like to thank all reviewers for your helpful feedback!

We add two tables for some additional experiments in response to your requests. Table 1 includes a new baseline for the MoJoCo task, as well as the omitted std of the proposed model. The proposed model still exhibits comparable performance to the baselines. Table 2 measures the time for prior and posterior sampling. It supports that importance sampling indeed improves efficiency.

---

> ### Comment · Area_Chair_bpGJ · 2023-08-18
>
> Dear authors,
>
> Thanks for your rebuttal. Unfortunately, not all reviewers have responded to your rebuttal yet. I just wanted to let you know that I'll keep trying to get the to reply to your rebuttal. In any case, the points raised in the rebuttal will be taken into account in the private reviewer-AC discussion and in the final decision making.
>
> -The AC

---

### Decision · Program_Chairs · 2023-09-21

**Decision:**

Accept (poster)

**Comment:**

The goal of the paper is to learn from demonstrations consisting of sequences of states only (without action labels), especially in non-Markovian settings. The authors propose an energy-based formulation that can be solved using inference technique. The resulting methods is tested in several domains with simulated robots, and shows good performance compared to a number of baselines.

One reviewer was critical of the paper, mentioning presentation, lack of baselines, performance in a stochastic environment, and relevance. The authors explained or rebutted most of these points with a clearly articulated rebuttal.

The other reviewers are mostly positive of the paper. The problem addressed is seen as relevant, novel, and challenging. The empirical results are also mostly seen as convincing. There was some concern about the learning objective, which was clarified in the discussion. Another suggestion was to add more examples of (realistic) nMDP problems and a more structured analysis in different types of non-Markovian tasks.